# R2DET: EXPLORING RELAXED ROTATION EQUIVARIANCE IN 2D OBJECT DETECTION

**Zhiqiang Wu**[1*]  **Yingjie Liu**[1*]   **Hanlin Dong**[1]   **Xuan Tang**[2]   **Jian Yang**[3]   **Bo Jin**[4]
**Mingsong Chen**[1]   **Xian Wei**[1†]

[1] Software Engineering Institute, East China Normal University
[2] School of Communication and Electronic Engineering, East China Normal University
[3] School of Geospatial Information, Information Engineering University
[4] School of Computer Science and Technology, Tongji University
51265902095@stu.ecnu.edu.cn   yjLiu6@hotmail.com   xian.wei@tum.de

## ABSTRACT

Group Equivariant Convolution (GConv) empowers models to explore underlying symmetry in data, improving performance. However, real-world scenarios often deviate from ideal symmetric systems caused by physical permutation, characterized by non-trivial actions of a symmetry group, resulting in asymmetries that affect the outputs, a phenomenon known as Symmetry Breaking. Traditional GConv-based methods are constrained by rigid operational rules within group space, assuming data remains strictly symmetry after limited group transformations. This limitation makes it difficult to adapt to Symmetry-Breaking and non-rigid transformations. Motivated by this, we mainly focus on a common scenario: Rotational Symmetry-Breaking. By relaxing strict group transformations within Strict Rotation-Equivariant group $\mathbf{C}_n$, we redefine a Relaxed Rotation-Equivariant group $\mathbf{R}_n$ and introduce a novel Relaxed Rotation-Equivariant GConv (R2GConv) with only a minimal increase of $4n$ parameters compared to GConv. Based on R2GConv, we propose a Relaxed Rotation-Equivariant Network (R2Net) as the backbone and develop a Relaxed Rotation-Equivariant Object Detector (R2Det) for 2D object detection. Experimental results demonstrate the effectiveness of the proposed R2GConv in natural image classification, and R2Det achieves excellent performance in 2D object detection with improved generalization capabilities and robustness. The code is available in `https://github.com/wuer5/r2det`.

## 1 INTRODUCTION

2D object detection is a crucial computer vision task with applications in various domains, including autonomous driving and geosciences (Zou et al., 2023; Kaur & Singh, 2023). Recent advancements in Deep Neural Networks (DNNs) (Hussain, 2023; Huang et al., 2017; Zagoruyko & Komodakis, 2016; Xie et al., 2017) have achieved remarkable progress. Nevertheless, objects within natural images often exhibit rotation and scale variations, requiring DNNs to handle geometric transformations more flexibly. One effective approach to address this issue is data augmentation, which improves object detection performance by rotating the dataset to expand additional data volume. Still, it leads to a considerable memory and training cost. Equivariant Neural Networks (ENNs) (Gerken et al., 2023) enhance feature learning and improve downstream task performance by incorporating symmetry and leveraging underlying physics. For instance, the Strict Rotation-Equivariance (SRE) introduced in Group Equivariant Convolution (GConv) (Cohen & Welling, 2016a) has demonstrated superior performance compared to traditional detectors (Han et al., 2021).

Using symmetry as an inductive bias in ENNs has emerged as a powerful tool, with significant conceptual and practical breakthroughs (Bronstein et al., 2021). Note that the symmetry refers to the fact that the input remains the same after transformation, rather than the observed symmetry of the object itself Weyl (1952). However, these ENNs merely or by default assume strict symmetry

---

*Equal Contribution
†Corresponding Author

across all features after equivariant transformations. Also, real-world data or features rarely conform to strict symmetry but often conform to non-strict symmetry after some transformations. Such non-strict symmetry is called Symmetry-Breaking, which means that features tend to shift from high symmetry to low symmetry in the real world.

As stated in Gross (1996), "*The secret of nature is symmetry, but much of the texture of the world is due to mechanisms of symmetry breaking.*"

Based on such natural law, we raises a critical question in ENNs as follows:

● *Are existing ENNs considering prior knowledge of such non-strict symmetry in real-world scenarios, such as 2D object detection and other computer vision tasks?*

***The answer is No.*** Following Curie's principle (Earman, 2004; Chalmers, 1970), equivariant functions preserve or enhance the symmetry of the input, making them inherently stricter. As a result, ENNs are unable to fully align with or adapt to non-strict symmetry present in real-world scenarios.

As stated in Curie's principle, the symmetry of a cause is always preserved in its effects, and any asymmetry in the effects must be present in their causes. This implies that the input with higher symmetry cannot be mapped to the output with lower symmetry since the input has no corresponding asymmetry. Conversely, the input with lower symmetry can be mapped to the output with higher symmetry. When an object experiences events such as rotations that exceed the predefined group's scope or the introduction of minor defects on its surface, it implies that the training images and the target object may have different levels of symmetry. In such cases, strict ENNs may encounter difficulties in accurately representing the object, strict adherence to symmetry constraints could prevent it from distinguishing between the object's perturbed and non-perturbed states, which is crucial for specific tasks. Some pioneer works (Locatello et al., 2020; Smidt et al., 2021; Wang et al., 2022a; Kaba & Ravanbakhsh, 2023; Huang et al., 2024; Xie & Smidt, 2024) discussed relaxation of equivariance and claimed that relaxed ENNs can model Symmetry-Breaking in multiple domains.

● *However, there is still a significant gap between these existing relaxed ENN works and practical applications of the computer vision field, e.g., 2D object detection.*

This work especially focuses on a common equivariance in ENNs, i.e., Rotation-Equivariance to address Rational Symmetry-Breaking. Differing from traditional SRE GConv, this work presents a novel Relaxed Rotation-Equivariance (RRE) GConv by incorporating a learnable adaptive rotational deviation parameter that is updated end-to-end by the dataset. Figure 1 details a case of Rotational Symmetry-Breaking and the proposed approach to solve it. RRE enhances the network's ability to recognize objects with relaxed rotational equivariance, effectively tackling the Symmetry-Breaking problem and capturing distinct features from perturbations, thus promoting performance. The main contributions of this work are as follows:

- We introduce RRE into the group convolution operation by simply incorporating learnable perturbations, proposing a Relaxed Rotation-Equivariant GConv (R2GConv).
- To our knowledge, we are the first to explore Rotational Symmetry-Breaking situations within vision tasks. We further propose a Relaxed Rotation-Equivariant Network (R2Net) as a backbone for better representation learning for Rotational Symmetry-Breaking.
- We redesign a Relaxed Rotation-Equivariant Object Detector (R2Det) for 2D object detection. Experimental results demonstrate that the proposed approach has achieved better convergence and outstanding performance with lightweight parameters.

## 2 RELATED WORKS

**Conv-based 2D Object Detectors.** 2D object detection (Chen et al., 2023; Zou et al., 2023; Kaur & Singh, 2023) is a fusion of object location and classification tasks, which involves locating objects through bounding boxes and identifying their respective categories. Several notable methods (Girshick et al., 2014; Liu et al., 2016; Carion et al., 2020; Wang et al., 2021; 2022b) have been developed, and the YOLO framework has stood out for its remarkable balance of speed and accuracy (Redmon et al., 2016; Hussain, 2023; Terven et al., 2023). Since its inception, the YOLO family has evolved through multiple iterations and variations, such as YOLOv8 (Jocher et al., 2023) YOLOv9 (Wang et al., 2024b), YOLOv10 (Wang et al., 2024a).

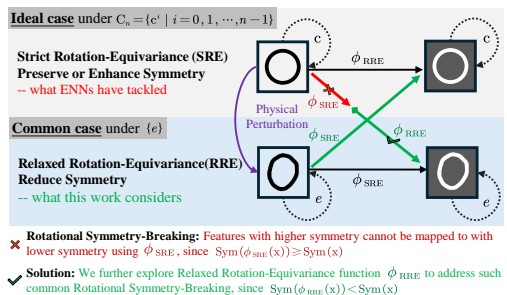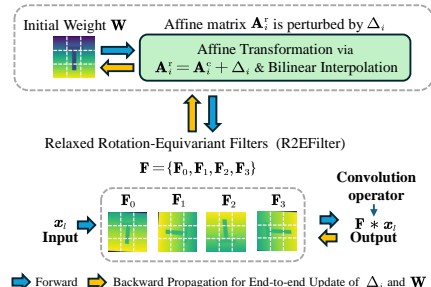

Figure 1: **Left:** The ideal feature of a circle with higher symmetry rarely occurs in real-world scenarios. Instead, the physical perturbation of higher symmetry results in lower symmetry. While ENNs can handle higher symmetry, Curie's principle dictates features of higher symmetry cannot be mapped to outputs with lower symmetry, inducing Symmetry-Breaking situations, and impairing feature learning. RRE function $\phi_{\text{RRE}}$ can solve Rotational Symmetry-Breaking situations, which has been proven by Kaba & Ravanbakhsh (2023). Note that $\text{Sym}(\cdot)$ denotes the level of its symmetry. **Right:** This work proposes the R2EFilter to build RRE, by incorporating learnable perturbation $\Delta$. We show the forward and backward processes of R2EFilter based on relaxed $\mathbf{C}_4$, named $\mathbf{R}_4$.

**Symmetry vs. Equivariance.** Symmetry is a fundamental characteristic where an object remains unchanged after certain transformations. Define the function $\text{Sym}(\cdot)$ denotes the level of its symmetry, and the equivariant network $\phi_{eq}$, we have $\text{Sym}(\phi_{eq}(x)) \geq \text{Sym}(x)$ based on Curie's principle. That means equivariance can preserve or enhance the symmetry of the input. In effect, equivariance can be used as a symmetry compiler to find missing symmetry-implied data.

**Common Equivariance: Rotation-Equivariance.** The concept of an equivariant network was proposed in Cohen & Welling (2016a), named Group Equivariant Convolution Neural Networks (G-CNN). Rotation-Equivariant convolution or full connect layer (Li et al., 2018; Marcos et al., 2017; Cohen & Welling, 2016a; Finzi et al., 2020) guarantees the rotation-equivariance of extracted features under the group operations by a higher degree of weight sharing. Moreover, Rotation-Equivariance has recently become a strongly desired prior bias in object detection tasks. Han et al. (2021) propose a rotation-equivariant 2D object detector (ReDet) to predict the orientation of aircraft accurately. Most recently, Lee et al. (2024) (FRED) achieves fully Rotation-Equivariant oriented object detection and enables more genuine non-axis-aligned learning. Wang et al. (2023) (DuEqNet) improves 3D object detection performance by constructing a dual-layer object detection network for 3D point clouds with rotation invariance and extracting local-global invariance features. Although Rotation-Equivariance has been considered, there is a naive assumption of uniform strict symmetry across all features, neglecting scenarios that require a relaxation of equivariant constraints.

**Symmetry-Breaking vs. Relaxed Equivariance**. Physical laws are governed by numerous symmetries and real-world data, such as complex datasets and graphs. However, it often deviates from strict mathematical symmetry due to noisy or incomplete data or inherent Symmetry-Breaking features in the underlying system. Several works aim to break the symmetry. Wang et al. (2022a) investigates approximately equivariant networks by incorporating relaxed weight sharing in group convolutions and weight-tying in steerable CNNs, respectively, thereby achieving a bias toward not strictly preserving symmetry. Kaba & Ravanbakhsh (2023) proposes a novel theoretical guidance for constructing relaxed equivariant multilayer perceptrons, going beyond the straightforward approach of adding noise to inputs and using an ENN (Locatello et al., 2020). Huang et al. (2024) tackles graph symmetry in real-world data by leveraging graph coarsening to establish approximate symmetries and propose a bias-variance tradeoff formula based on symmetry group selection. Xie & Smidt (2024) introduces Symmetry-Breaking parameters sampled as model inputs from a set determined solely by input and output symmetries. These works further observe that breaking more symmetry than needed is beneficial.

In a nutshell, real-world data rarely conforms to strict symmetry. Diverse objects inherently possess levels of relaxed symmetry, regardless of modality, e.g., 2D images or 3D vision data (Kaba & Ravanbakhsh, 2023). Consequently, existing methods fail to tackle Symmetry-Breaking, affecting downstream task performance. To our knowledge, we are the first to address these limitations in the 2D object detection field by adopting the Rotational Symmetry-Breaking and RRE perspective.

## 3 PROPOSED METHOD

### 3.1 PRELIMINARY

**Definition 1 (*Strict Equivariance*).** *A learning function $\phi_{strict} : X \to Y$ that sends elements from input space $X$ to output space $Y$ satisfies Strict Equivariance to a group $G$ if $\forall g, \mathbf{x} \in G \times X$ there exists $\rho_X : G \to \mathrm{GL}(X)$ and $\rho_Y : G \to \mathrm{GL}(Y)$ actions of $G$ such that*

$$\phi_{strict}(\rho_X(g) \cdot \mathbf{x}) = \rho_Y(g) \cdot \phi_{strict}(\mathbf{x}), \tag{1}$$

*where $\mathrm{GL}(\cdot)$ is a general linear group over the space.*

**Definition 2 (*Relaxed Equivariance*).** (Kaba & Ravanbakhsh, 2023) *A learning function $\phi_{relaxed}$ : $X \to Y$ that sends elements from input space $X$ to output space $Y$ satisfies Relaxed Equivariance to a group $G$ if $\forall g_1, \mathbf{x} \in G \times X$ there exists $g_2 \in g_1 G_{\mathbf{x}}, \rho_X : G \to \mathrm{GL}(X)$ and $\rho_Y : G \to \mathrm{GL}(Y)$ actions of $G$ such that*

$$\phi_{relaxed}(\rho_X(g_1) \cdot \mathbf{x}) = \rho_Y(g_2) \cdot \phi_{relaxed}(\mathbf{x}), \tag{2}$$

**Strict Equivariant Neural Networks**. Learning equivariant features is an optimization process for a series of $\phi_{strict}$ function sets in the model. However, the challenge of strict equivariant networks lies in designing trainable layers, such as equivariant convolutions. Usually, there are two strategies for designing equivariant convolutions: weight sharing and weight typing, which are G-CNN and G-steerable CNN (Cohen & Welling, 2016b), respectively.

**Relaxed Equivariant Neural Networks**. The existing equivariant networks assume that the data is completely symmetric. This network approximates a strict invariant or equivariant function under given group actions. For example, in G-CNN, the shared convolution filters achieve equivariant images at 0, 90, 180, and 270 degrees under the strict constraint of $\mathbf{C}_4$. However, real-world data is rarely symmetric. This seriously hinders the potential application of equivariant networks. To solve this problem, in Elsayed et al. (2020) and Wang et al. (2022a), relaxing weight constraints can significantly improve the performance and generalization ability of the model. This work relaxes strict group constraints for a relaxed equivariant neural network.

### 3.2 RELAXED ROTATION-EQUIVARIANT FILTER (R2EFILTER)

In the following, we denote all $i \in \{0, 1, \cdots, n-1\}$ for convenient presentation. We first introduce Relaxed Rotation-Equivariant Filter (R2EFilter) denoted by **F**, serves as the crucial component of R2GConv. To relax strict group constraints, R2EFilter embeds a learnable parameter $\Delta$ to perturb the group operation based on the $n$-order cyclic rotation group $\mathbf{C}_n = \{\mathbf{c}^i \mid i = 0, 1, \cdots, n - 1\}$, a discrete and finite subgroup of SO(2), which an infinite group that contains a set of all two-dimensional rotation angles. The powers of $\mathbf{c}^i$ indicates performing rotation operation on the input $\mathbf{x}$ by $\frac{2\pi i}{n}$ degrees, $i$ times. The affine matrix $\mathbf{A}_i^{\mathbf{c}}$ on $\mathbf{C}_n$ can be defined as follows:

$$\mathbf{A}_i^{\mathbf{c}} = \begin{bmatrix} \cos(2\pi i/n) & -\sin(2\pi i/n) \\ \sin(2\pi i/n) & \cos(2\pi i/n) \end{bmatrix}. \tag{3}$$

Further, let the learnable perturbation factor $\Delta \in \mathbb{R}^{n \times 2 \times 2} \leftarrow \mathcal{U}(-b, b)$, where $\mathcal{U}$ denotes Uniform distribution with boundary value $b$. Our original intention is to provide an appropriate level of initial perturbation to fit $\Delta$ to a suitable value. In Section 4, we conduct several experiments with different $b$ from 0 to 0.8, where the experimental results show that a small value of $b$ enhances the model's performance, whereas a large value of $b$ detrimentally affects the model's performance.

Then, we can define $\Delta = \{\Delta_i\}, \Delta_i = [\Delta_{i1}, \Delta_{i2}, \Delta_{i3}, \Delta_{i4}]$ and a Relaxed Rotation-Equivariant group $\mathbf{R}_n = \{\mathbf{r}^i\}$ based on $\mathbf{C}_n$ and a transformation function $\mathcal{T} : \mathbf{C}_n \to \mathbf{R}_n$. We consider $\mathcal{T}$ as an addition, i.e., adding the learnable perturbation factor $\Delta_i$ to the affine matrix $\mathbf{A}_i^{\mathbf{c}}$. Thus, the perturbed affine matrix $\mathbf{A}_i^{\mathbf{r}}$ on $\mathbf{R}_n$ is defined as $\mathbf{A}_i^{\mathbf{r}} = \mathcal{T}(\mathbf{A}_i^{\mathbf{c}}, \Delta_i) = \mathbf{A}_i^{\mathbf{c}} + \Delta_i$, where other operations are also available for $\mathcal{T}$, such as multiplication, and linear transformation, but are not limited to.

Here, we explain the affine transformation from the perspective of coordinate changes. Specifically, given **W** as a Kaiming-Initialization (He et al., 2015) 2D convolution weight, and **F** denotes the transformed filter on $\mathbf{R}_n$. Consider this example, let a function CoorSet($\cdot$) denote a set of 2D coordinates of the elements in $\cdot$, where the coordinate system is at the center of $\cdot$. For all coordinates

$[u\,v] \in \texttt{CoorSet}(\mathbf{W})$, we have two affine transformations: $\mathbf{c}^i([u\,v]) = \mathbf{A}_i^{\mathbf{c}} \cdot [u\,v]^\top$ and $\mathbf{r}^i([u\,v]) = \mathbf{A}_i^{\mathbf{r}} \cdot [u\,v]^\top$ on $\mathbf{C}_n$ and $\mathbf{R}_n$, respectively. The transformed coordinates $[\tilde{u}_i\,\tilde{v}_i]$ for $\mathbf{F}_i$ are as follows:

$$\mathbf{r}^i([u\,v]) = \left( \begin{bmatrix} \cos\left(2\pi i/n\right) + \Delta_{i1} & -\sin\left(2\pi i/n\right) + \Delta_{i2} \\ \sin\left(2\pi i/n\right) + \Delta_{i3} & \cos\left(2\pi i/n\right) + \Delta_{i4} \end{bmatrix} \begin{bmatrix} u \\ v \end{bmatrix} \right), \quad [\tilde{u}_i\,\tilde{v}_i] = \lfloor \left( \mathbf{r}^i([u\,v]) \right)^\top \rfloor, \quad (4)$$

where $\lfloor \cdot \rfloor$ denotes the round down operation for $\cdot$. Further, if $[\tilde{u}_i\,\tilde{v}_i] \in \texttt{CoorSet}(\mathbf{F}_i)$, we set $\mathbf{F}_i[\cdots, \tilde{u}_i, \tilde{v}_i] := \mathbf{W}[\cdots, u, v]$. For coordinates $[\dot{u}_i\,\dot{v}_i]$ in $\mathbf{F}_i$ that remain unassigned after mapping from $\mathbf{W}$, we employ the ***Bilinear Interpolation*** method to fill $\mathbf{F}_i[\cdots, \dot{u}_i, \dot{v}_i]$. Then, we have the complete R2EFilter: $\mathbf{F} = \{\mathbf{F}_i\}$. The construction process of $\mathbf{F}$ is detailed in Algorithm 1. In the rest of this work, this process is abstracted by the mapping function $\Psi$, i.e., $\mathbf{F} = \Psi(\mathbf{W})$.

---

**Algorithm 1** Build R2EFilter based on $\mathbf{C}_n$.

---

**Input:** The group order $n$ of $\mathbf{C}_n$; The boundary value $b$ of Uniform distribution $\mathcal{U}(-b, b)$; The Kaiming-Initialization $\mathbf{W}$.
**Output:** The R2EFilter based on $\mathbf{C}_n$: $\mathbf{F}$.
Initialize $\mathbf{F} \leftarrow \emptyset$              ▷ Empty $\mathbf{F}$ to store compelete R2EFilter
Initialize $\Delta \in \mathbb{R}^{n \times 2 \times 2} \leftarrow \mathcal{U}(-b, b)$    ▷ Learnable perturbation factor with Uniform distribution
$s \leftarrow \texttt{Get\_Tensor\_Shape}(\mathbf{W})$           ▷ Get the tensor shape
$c \leftarrow \texttt{Get\_Output\_Channels}(\mathbf{W})$         ▷ Get the output channels
**for** $i = 0$ **to** $(n-1)$ **do**             ▷ Loop the group order $n$
  $\mathbf{A}_i \leftarrow \begin{bmatrix} \cos(2\pi i/n) + \Delta_{i1} & -\sin(2\pi i/n) + \Delta_{i2} & 0 \\ \sin(2\pi i/n) + \Delta_{i3} & \cos(2\pi i/n) + \Delta_{i4} & 0 \end{bmatrix}$   ▷ Get the perturbed affine matrix
  $\mathbf{A}_i^c \leftarrow \texttt{Repeat\_Tensor}(\mathbf{A}_i, c)$         ▷ Repeat $\mathbf{A}_i$ for $c$ times
  $\mathbf{G}_i \leftarrow \texttt{Affine\_Grid}(\mathbf{A}_i^c, s)$         ▷ Generate affine grid $\mathbf{G}_i$
  $\mathbf{F}_i \leftarrow \texttt{Grid\_Sample}(\mathbf{W}, \mathbf{G}_i)$     ▷ Sampling on $\mathbf{G}_i$ to get the $i$-order filter $\mathbf{F}_i$
  Update $\mathbf{F} \leftarrow \mathbf{F} \cup \mathbf{F}_i$
**end for**
**return** $\mathbf{F}$

---

### 3.3 RELAXED ROTATION-EQUIVARIANT GCONV (R2GCONV)

So far, by relaxing strict constraints of group operations on $\mathbf{C}_n$ through learnable perturbation factor $\Delta$, we have defined the Relaxed Rotation-Equivariant group $\mathbf{R}_n$ and achieved R2EFilter. Based on it, we introduce a Relaxed Rotation-Equivariant GConv (R2GConv), encompassing three variants: Lifting R2GConv, Point-wise R2GConv, and Depth-wise R2GConv.

Specifically, since the input image and traditional convolution filters are on the plane, Lifting R2GConv is designed as an equivariant transformation to replace conventional translation operations. It is applied in the first layer to convert the input data into feature maps on a specific group (e.g., $\mathbf{R}_n$ in this work), enabling more complex and structured feature extraction. For subsequent layers, filters of Point-wise R2GConv and Depth-wise R2GConv are defined on $\mathbf{R}_n$, facilitating the equivariant transformation and processing of these feature maps. The point-wise and depth-wise strategies in R2GConv avoid the inevitably substantial parameters and computational overhead associated with GConv operations. Meanwhile, the reduced number of parameters also diminishes the computational complexity and required time to build their filters with R2EFilter. Based on this reason, we combine Point-wise R2GConv and Depthwise R2GConv to propose an Efficient R2GConv.

For convenience, in the following, let $n_l$, $c_l$, $c_{l+1}$, $k_l$, $h_l$, and $w_l$ denote the group order, the number of input channels, the number of output channels, convolution kernel size, input height, and input width in the $l$-layer, respectively.

● **R2GConv.** Consider the input feature map $\boldsymbol{x}_l \in \mathbb{R}^{c_l n_l \times h_l \times w_l}$ and an initial convolution weight $\mathbf{W}_l \in \mathbb{R}^{c_{l+1} \times c_l n_l \times k_l \times k_l}$, where $l > 1$. Traditional convolution computes the output feature map on a plane by performing an inner product between the input feature map and the convolution filter, with the filter shifted by a defined step. According to Section 3.2, we have R2EFilter $\mathbf{F}_l = \Psi(\mathbf{W}_l) \in \mathbb{R}^{c_{l+1} n_{l+1} \times c_l n_l \times k_l \times k_l}$, where $n_{l+1} = n_l = n$. Thus R2GConv transforms the input feature map $\boldsymbol{x}_l$ to the output feature map $\boldsymbol{x}_{l+1} \in \mathbb{R}^{c_{l+1} n_{l+1} \times h_{l+1} \times w_{l+1}}$ as follows:

$$\underbrace{\boldsymbol{x}_{l+1}[d, :, :]}_{h_{l+1} \times w_{l+1}} = \sum_{c=0}^{c_l n_l - 1} \underbrace{\mathbf{F}_l[d, c, :, :]}_{k_l \times k_l} * \underbrace{\boldsymbol{x}_l[c, :, :]}_{h_l \times w_l}, \quad \forall d \in \{0, \cdots, c_{l+1} n_{l+1} - 1\}, \quad (5)$$

where $*$ denotes vanilla convolution operation. We also prove that our R2GConv is the `RRE` block, as shown in Appendix 7.10.

• **Lifting R2GConv.** Considering the special case where R2GConv is employed in the first layer, i.e., $l = 1$, to lift the input feature map $\boldsymbol{x}_1 \in \mathbb{R}^{c_1 \times h_1 \times w_1}$ onto $\mathbf{R}_n$, called Lifting R2GConv. Thus, we have its R2EFilter $\bar{\mathbf{F}}_1 = \Psi(\mathbf{W}_1) \in \mathbb{R}^{c_2 n_2 \times c_1 n_1 \times k_1 \times k_1}$, where $n_1 = 1, n_2 = n$. Further, the output feature map $\boldsymbol{x}_2 \in \mathbb{R}^{c_2 n_2 \times h_2 \times w_2}$ is obtained as follows:

$$\underbrace{\boldsymbol{x}_2[d,:,:]}_{h_2 \times w_2} = \sum_{c=0}^{c_1-1} \underbrace{\mathbf{F}_1[d,c,:,:]}_{k_1 \times k_1} * \underbrace{\boldsymbol{x}_1[c,:,:]}_{h_1 \times w_1}, \quad \forall d \in \{0, \cdots, c_2 n_2 - 1\}. \tag{6}$$

• **Point-wise R2GConv.** In the $l$-layer with $l > 1$, given the input feature map $\boldsymbol{x}_l \in \mathbb{R}^{c_l n_l \times h_l \times w_l}$ and an initial convolution weight $\mathbf{W}_l^{\mathrm{p}} \in \mathbb{R}^{c_{l+1} \times c_l n_l \times 1 \times 1}$, we have $\mathbf{F}_l^{\mathrm{p}} = \Psi(\mathbf{W}_l^{\mathrm{p}}) \in \mathbb{R}^{c_{l+1} n_{l+1} \times c_l n_l \times 1 \times 1}$, where $n_l = n_{l+1} = n$. Since the size of filter $\mathbf{F}_l^{\mathrm{p}}$ is always $1 \times 1$, the Point-wise R2GConv is considered the standard point-wise convolution operation, which is defined as follows:

$$\underbrace{\boldsymbol{x}_{l+1}[d,:,:]}_{h_{l+1} \times w_{l+1}} = \sum_{c=0}^{c_l n-1} \underbrace{\mathbf{F}_l^{\mathrm{p}}[d,c,:,:]}_{1 \times 1} * \underbrace{\boldsymbol{x}_l[c,:,:]}_{h_l \times w_l}, \quad \forall d \in \{0, \cdots, c_{l+1} n_{l+1} - 1\}. \tag{7}$$

• **Depth-wise R2GConv.** Given the input feature map $\boldsymbol{x}_l \in \mathbb{R}^{c_l n_l \times h_l \times w_l}$ and an initial convolution weight $\mathbf{W}_l^{\mathrm{d}} \in \mathbb{R}^{c_{l+1} \times 1 \times k_l \times k_l}$ in the $l$-layer with $l > 1$, we have $\mathbf{F}_l^{\mathrm{d}} = \Psi(\mathbf{W}_l^{\mathrm{d}}) \in \mathbb{R}^{c_{l+1} n_{l+1} \times 1 \times k_l \times k_l}$, where $n_l = n_{l+1} = n$. Consequently, Depth-wise R2GConv is defined as follows:

$$\underbrace{\boldsymbol{x}_{l+1}[d,:,:]}_{h_{l+1} \times w_{l+1}} = \underbrace{\mathbf{F}_l^{\mathrm{d}}[d,:,:,:]}_{1 \times k_l \times k_l} *_{\mathrm{D}} \underbrace{\boldsymbol{x}_l[:,:,:]}_{c_l n_l \times h_l \times w_l}, \quad \forall d \in \{0, \cdots, c_{l+1} n_{l+1} - 1\}, \tag{8}$$

where $*_{\mathrm{D}}$ denotes the depth-wise operation with the convolution group number $c_{l+1} n_{l+1}$, which needs to meet $c_l / c_{l+1} \in \mathbb{Z}^+$ for the correct division of convolution group number.

• **Efficient R2GConv.** Here, we propose an Efficient R2GConv as $\boldsymbol{x}_{l+1} = \mathbf{F}_l^{\mathrm{d}} *_{\mathrm{D}} (\mathbf{F}_l^{\mathrm{p}} * \boldsymbol{x}_l)$ in the $l$-layer, where $n_l = n_{l+1} = n$. Note that the point-wise operation is used for channel connection, while the depth-wise operation reduces the high parameters and computation. With these operations, the model's performance can be effectively maintained while achieving parameter reduction.

## 3.4 THE RELAXED ROTATION-EQUIVARIANT NETWORK (R2NET)

Based on R2GConv, we further propose a Relaxed Rotation-Equivariant Network (R2Net), as shown in Figure 2. R2Net comprises a Lifting R2GConv and the standard four-stage processing used for most backbone networks. In the first layer, we typically project the input tensor to our defined group $\mathbf{R}_n$ while performing $2\times$ downsampling by a Lifting R2GConv with stride 2. Then, we input the projected tensor into the four-stage processing. Each stage incorporates an Efficient R2GConv with stride 2 for $2\times$ downsampling, followed by an R2Net Block for feature extraction. In the last stage, we maintain the output channels unchanged compared to the input channels, which mainly reduces the number of parameters. Regarding R2Net Block, we borrow the idea of dividing channels in Res2Net (Gao et al., 2019) to reduce the number of parameters and FLOPs. In the R2Net Block, we initially apply a Point-wise R2GConv for the channel change and then split the channel. Subsequently, for each channel after the first one, we aggregate it with the previous channel before feeding it into its corresponding Bottleneck, which is composed of two stacked Efficient R2GConv modules with a residual connection (He et al., 2016) structure to improve the convergence ability.

Here, we provide our R2Net with three sizes: R2Net-N, R2Net-S, and R2Net-M, with the suffixes "-N", "-S" and "-M" indicating the Nano size (as introduced above), Small size, and Medium size, respectively. Detailed parameter settings of R2Net can be found in Appendix 7.1.

## 3.5 THE REDESIGNED RELAXED ROTATION-EQUIVARIANT OBJECT DETECTOR (R2DET)

Based on the backbone R2Net, we propose a novel Relaxed Rotation-Equivariant Object Detector (R2Det). R2Det employs the Feature Pyramid Network (FPN) + Path Aggregation Network (PAN) neck architecture, as shown in Figure 3. Specifically, R2Net mainly extracts multi-scale features

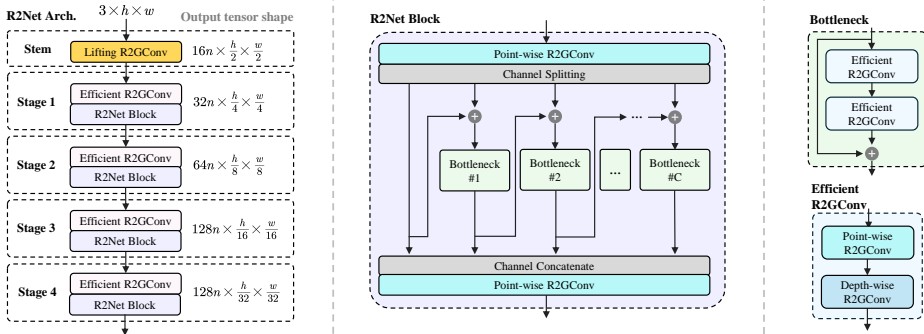

Figure 2: The architecture of R2Net-N as the backbone for feature extraction, where #C denotes the number of Bottlenecks in R2Net Blocks based on the channel sizes and varies with the different sizes of R2Net (i.e., R2Det-N / S / M). Note that all R2GConv including three variants have normalization (BatchNorm) and activation (SiLU) functions, which we do not show in the figure.

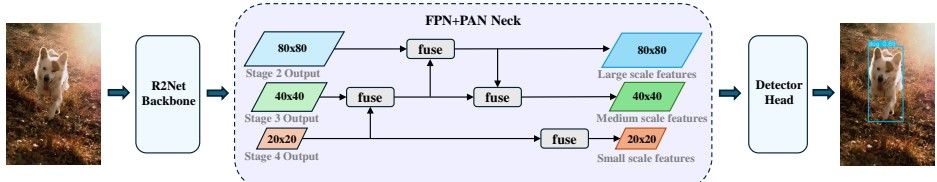

Figure 3: The architecture of R2Det for 2D object detection with a FPN+PAN neck. Here, we only show a simple architectural diagram. For detailed architecture, please refer to Appendix 7.2.

in R2Det. The neck part in R2Det further aggregates information from multiple paths, allowing for enhanced communication between different levels of the feature pyramid. Following three sizes of R2Net, we also provide our R2Det with three sizes: R2Det-N, R2Det-S, and R2Det-M. More detailed parameter settings of R2Det can be found in Appendix 7.1.

## 4 EXPERIMENTS

### 4.1 2D OBJECT DETECTION ON THE PASCAL VOC AND MS COCO DATASETS

To investigate the effectiveness of our method, we conduct extensive experiments on the MSCOCO 2017 (COCO) and PASCAL VOC07+12 (VOC) datasets. We set $b = 0.1$, and the RRE (or SRE) is constructed on $\mathbf{C}_4$ in all experiments if not specified.

**Effect of $b$ in Uniform distribution $\mathcal{U}(-b, b)$ on RRE.** We explore the impact of the boundary value $b$ of uniform distribution $\mathcal{U}(-b, b)$ for the perturbation on model performance, where a larger value of $b$ indicates a greater initial perturbation $\Delta$ and has a higher level of RRE in R2GConv. Table 1 demonstrate that R2Det-N with $b = 0.1$ achieve the best results in both $\mathbf{AP}_{50:95}$ and $\mathbf{AP}_{50}$, outperforming those with initializations of 0 or too large values. Figure 4a further demonstrates the training curves of our R2Det-N ($\mathbf{C}_4$) with varying $b$ on VOC dataset. Since the $\Delta$ in R2GConv can be readjusted by end-to-end learning strategy, we suggest that a small initial perturbation can better align $\Delta$ to the Rotational Symmetry-Breaking property of natural image datasets. Additionally, we provide specific $\Delta$ values in four Efficient R2GConv, as shown in Appendix 7.4. Note that R2Det (RRE) with $b = 0$ is not equivalent to R2Det (SRE), as $\Delta$ with an initial value of 0 still achieves end-to-end updates by gradient descent.

**Convergence analysis.** We analyze the convergence of our R2Det and YOLOv8 models during the training process on the VOC training dataset, as shown in Figure 4b. Both R2Det-S / M converge in about 66 epochs, much earlier than YOLOv8-S / M, which converge in about 138 epochs and 132 epochs, respectively. Furthermore, YOLOv8-N converges in about 198 epochs, while R2Det-N converges in about 132 epochs, the same as YOLOv8-M. It is worth noting that the convergence curve of R2Det is smoother than YOLOv8, demonstrating a better convergence process of R2Det.

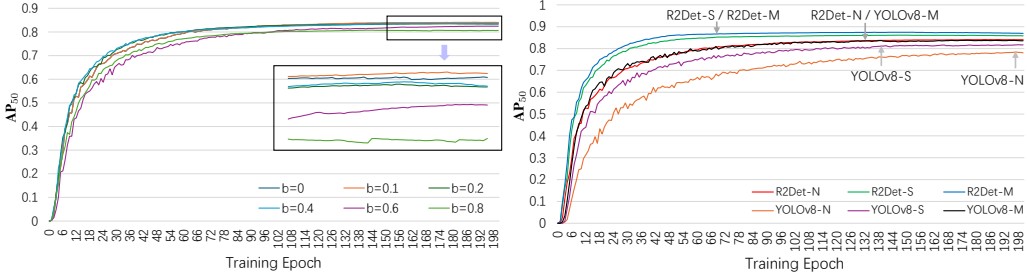

(a) $\mathbf{AP}_{50}$ of R2Det with different $b$      (b) $\mathbf{AP}_{50}$ comparison of R2Det and YOLOv8.

Figure 4: $\mathbf{AP}_{50}$ curves on VOC test dataset. All models train for 200 epochs with the same settings.

Overall, our R2Det models of all three variants (N / S / M) exhibit not only outstanding performance in $\mathbf{AP}_{50}$ but also a more stable training process with faster convergence speed compared to YOLOv8 models. The faster convergence could be related to the fact that our R2GConv can extract rich and relaxed equivariant features, enabling earlier learning of these potential features.

Table 1: Results of R2Det-N ($\mathbf{C}_4$) with different boundary value $b$ on VOC test dataset.

| $b$ | $\mathbf{AP}_{50}(\%)$ | $\mathbf{AP}_{50:95}(\%)$ |
|---|---|---|
| 0 | 83.8 | 64.4 |
| 0.1 | **84.1** | **65.1** |
| 0.2 | 83.5 | 64.3 |
| 0.4 | 83.6 | 64.4 |
| 0.6 | 82.4 | 62.6 |
| 0.8 | 80.7 | 59.7 |

Table 2: Ablation experiments of R2Det-N with SRE and RRE (Ours) on VOC test dataset.

| Group | Equiv. | $\mathbf{AP}_{50:95}(\%)$ | $\mathbf{AP}_{50}(\%)$ | Params. | FLOPs |
|---|---|---|---|---|---|
| $\mathbf{C}_2$ | SRE | 58.4 | 78.1 | 1.8M | 3.2G |
| | RRE | **59.0** (+0.6) | **79.4** (+1.3) | 1.8M + 0.256K | 3.2G |
| $\mathbf{C}_4$ | SRE | 64.2 | 82.9 | 2.6M | 3.3G |
| | RRE | **65.1** (+0.9) | **84.1** (+1.2) | 2.6M + 0.512K | 3.3G |
| $\mathbf{C}_8$ | SRE | 65.5 | 84.2 | 4.4M | 3.7G |
| | RRE | **67.0** (+1.5) | **85.4** (+1.2) | 4.4M + 1.024K | 3.7G |

Table 3: Ablation experiments of R2Det-N with SRE and RRE (Ours) on COCO validation dataset.

| Group | Equiv. | $\mathbf{AP}_{50:95}(\%)$ | $\mathbf{AP}_{75}(\%)$ | $\mathbf{AP}_{50}(\%)$ | $\mathbf{AP}_S(\%)$ | $\mathbf{AP}_M(\%)$ | $\mathbf{AP}_L(\%)$ | Params. | FLOPs |
|---|---|---|---|---|---|---|---|---|---|
| $\mathbf{C}_2$ | SRE | 36.2 | 39.5 | 51.0 | 18.2 | 39.8 | 50.7 | 1.9M | 3.8G |
| | RRE | **36.6** (+0.4) | **39.8** (+0.3) | **51.5** (+0.5) | **19.4** (+1.2) | **39.9** (+0.1) | **51.5** (+0.8) | 1.9M + 0.256K | 3.8G |
| $\mathbf{C}_4$ | SRE | 43.2 | 47.4 | 58.7 | 22.6 | 48.1 | 58.3 | 2.8M | 4.0G |
| | RRE | **43.7** (+0.5) | **47.8** (+0.4) | **59.5** (+0.8) | **24.2** (+1.6) | **48.2** (+0.1) | **59.3** (+1.0) | 2.8M + 0.512K | 4.0G |
| $\mathbf{C}_8$ | SRE | 46.8 | 62.2 | 51.0 | 26.0 | 52.7 | 62.1 | 4.5M | 4.3G |
| | RRE | **47.5** (+0.7) | **63.7** (+0.5) | **52.1** (+1.1) | **27.9** (+1.9) | **52.9** (+0.2) | **63.4** (+1.3) | 4.5M + 1.024K | 4.3G |

• Note that "Equiv." denotes the type of rotation-equivariant filter used in R2Det, as SRE indicates regular Strict Rotation-Equivariant Filter, whereas RRE means our proposed Relaxed Rotation-Equivariant Filter (R2EFilter).

**Ablation experiments evaluating RRE vs. SRE on COCO and VOC.** We first conduct ablation experiments on COCO and VOC datasets to discuss the effectiveness of RRE, comparing it with SRE, based on three groups, including $\mathbf{C}_2$, $\mathbf{C}_4$, and $\mathbf{C}_8$. As shown in Table 2 and Table 3, R2Det-N (RRE) outperforms R2Det-N (SRE) in all average precision (**AP**) metrics. These results demonstrate that RRE can effectively overcome the limitations of SRE mentioned above. Meanwhile, compared to R2Det-N (SRE), our approach results in a negligible increase in parameters quantity (**Params.**) and floating point operations (**FLOPs**) compared to R2Det-N (SRE), improving model performance with **almost negligible** parameter increase (i.e., 256 / 512 / 1024) on $\mathbf{C}_2$ / $\mathbf{C}_4$ / $\mathbf{C}_8$.

**Inability of naive convolutions with learnable noise to construct RRE.** We conduct experiments that incorporate learnable noise into the traditional convolution filters of YOLOv8 to evaluate its impact on convolution filters. As shown in Table 4, simply adding noise to the filters of the original convolution leads to degraded model performance. This suggests that for operators like the original convolution, which lack inherent rotation-equivariance properties, simply adding noise does not confer achieving RRE to handle data exhibiting Rotational Symmetry-Breaking.

Table 4: Ablation experiments of naive convolution filter w/ (w/o) learnable noise of YOLOv8-N on VOC test dataset.

| Type | $\mathbf{AP}_{50}(\%)$ | $\mathbf{AP}_{50:95}(\%)$ |
|---|---|---|
| **w/o** noise | 78.6 | 57.5 |
| **w/** noise | 73.1 (-5.5) | 51.4 (-6.1) |

Table 5: Ablation experiments of YOLOv8-N architecture on $\mathbf{C}_4$ on VOC test dataset.

| Equiv. | $\mathbf{AP}_{50:95}(\%)$ | $\mathbf{AP}_{50}(\%)$ | Params. |
|---|---|---|---|
| NRE | 57.5 | 78.6 | 3.3M |
| SRE | 62.8 | 82.7 | 3.2M |
| RRE | **64.2** (+1.4) | **83.6** (+0.9) | 3.2M |

**Plug-and-play and generalization ability.** We show that by simply replacing and plugging, our proposed modules mentioned in Section 3.3 can effectively transfer the RRE property to other architectures, such as YOLOv8 architecture. Specifically, we replace the first layer Conv of YOLOv8-N with Lifting R2GConv and replace all other intermediate layer Convs with our Efficient R2GConv. From Table 5, YOLOv8-N (RRE) also surpass YOLOv8-N (SRE) and YOLOv8-N (NRE) in both $AP_{50:95}$ and $AP_{50}$. Note that YOLOv8-N (NRE) means the standard YOLOv8-N model. This experiment demonstrates that our R2GConv is plug-and-play and still effective in other architectures, demonstrating its generalization ability.

Table 6: Comparison with the latest state-of-the-art object detectors on COCO validation dataset.

| Method | $AP_{50:95}(\%)$ | $AP_{50}(\%)$ | $AP_{75}(\%)$ | $AP_S(\%)$ | $AP_M(\%)$ | $AP_L(\%)$ | Params. | FLOPs |
|---|---|---|---|---|---|---|---|---|
| YOLOv8-N | 37.3 | 52.6 | 40.5 | 18.6 | 41.0 | 53.5 | 3.2M | 8.7G |
| YOLOv8-S | 44.9 | 61.8 | 48.7 | 26.0 | 49.9 | 61.1 | 11.2M | 28.6G |
| YOLOv8-M | 50.2 | 67.2 | 54.7 | 32.3 | 55.9 | 66.5 | 25.9M | 78.9G |
| YOLOv8-L | 52.9 | 69.8 | 57.5 | 35.3 | 58.3 | 69.8 | 43.7M | 165.2G |
| YOLOv9-T | 38.3 | 53.1 | 41.3 | - | - | - | 2.0M | 7.7G |
| YOLOv9-S | 46.8 | 63.4 | 50.7 | 26.6 | 56.0 | 64.5 | 7.1M | 26.4G |
| YOLOv9-M | 51.4 | 68.1 | 56.1 | 33.6 | 57.0 | 68.0 | 20.0M | 76.3G |
| YOLOv9-C | 53.0 | 70.2 | 57.8 | 36.2 | 58.5 | 69.3 | 25.3M | 102.1G |
| YOLOv10-N | 38.5 | 53.8 | 41.7 | 18.9 | 42.4 | 54.6 | 2.3M | 6.7G |
| YOLOv10-S | 46.3 | 63.0 | 50.4 | 26.9 | 51.1 | 63.7 | 7.2M | 21.6G |
| YOLOv10-M | 51.1 | 68.1 | 55.8 | 33.8 | 56.5 | 67.0 | 15.4M | 59.1G |
| YOLOv10-B | 52.5 | 69.6 | 57.2 | 35.1 | 57.8 | 68.4 | 19.1M | 92.0G |
| YOLOv10-L | 53.2 | 70.1 | 58.0 | 35.7 | 58.4 | 69.4 | 24.4M | 120.3G |
| **R2Det-N** ($C_4$) | 43.7 | 59.5 | 47.8 | 24.2 | 48.2 | 59.3 | 2.8M | 4.0G |
| **R2Det-S** ($C_4$) | 50.0 | 66.5 | 54.6 | 30.5 | 55.7 | 66.2 | 9.6M | 9.0G |
| **R2Det-M** ($C_4$) | 53.1 | 70.3 | 57.9 | 36.4 | 58.7 | 69.6 | 22.6M | 17.3G |

**Average precision comparison.** As detailed in Table 6, we compare our proposed R2Det with advanced YOLO-serie detectors on COCO validation dataset. Since both YOLOv8 and R2Det offer N / S / M variants, each of which is compared individually. Despite having 12.5% / 14.3% / 12.7% fewer parameters, R2Det achieves significant improvements of 17.2% / 11.4% / 5.8% in $AP_{50:95}$, compared to YOLOv8, respectively. Then, we compare R2Det to the latest advanced models. Also on N / S / M three variants, although these models have fewer parameters than R2Det, we achieve $AP_{50:95}$ improvements by 14.1% / 6.8% / 3.3% compared to YOLOv9, and 13.5% / 8% / 3.9% compared to YOLOv10. Moreover, while R2Det-M has approximately half the number of parameters of YOLOv8-L and lower FLOPs than YOLOv9-C and YOLOv10-L, it consistently outperforms YOLOv8-L, YOLOv9-C, and YOLOv10-B in **AP** across all IoU thresholds and for objects of different sizes. In addition, R2Det-M achieves approximate results compared to YOLOv10-L, but its parameters are fewer. We further conduct experiments on VOC dataset as shown in Table 7. In each variant comparison, R2Det exceeds YOLOv8 in $AP_{50}$ by 13.1% / 7.6% / 4.6%, for the N, S, and M variants, respectively. R2Det-M achieves state-of-the-art performance in $AP_{50}$ on VOC with only one-third parameters of YOLOv8-L. Experimental results above show that our R2Det has achieved excellent performance in **AP** with fewer parameters and much lower FLOPs than existing state-of-the-art object detectors. Please refer to Appendix 7.12 for more results and comparisons.

Table 7: Comparison with advanced YOLOv8 models on VOC test dataset.

| Method | $AP_{50}(\%)$ | Params. | FLOPs |
|---|---|---|---|
| YOLOv8-N | 78.6 | 3.0M | 8.1G |
| YOLOv8-S | 81.6 | 11.1M | 28.5G |
| YOLOv8-M | 83.7 | 25.9M | 78.7G |
| YOLOv8-L | 86.4 | 43.6M | 164.9G |
| YOLOv8-X | 86.9 | 68.1M | 257.5G |
| **R2Det-N** ($C_4$) | 84.1 | 2.6M | 3.3G |
| **R2Det-S** ($C_4$) | 86.0 | 9.6M | 8.9G |
| **R2Det-M** ($C_4$) | **87.3** | 22.6M | 17.2G |

Table 8: Comparison with other models in Top-1 Accuracy (%) on CIFAR-10 / 100 datasets.

| Method | C-10(%) | C-100(%) | Params. |
|---|---|---|---|
| WideResNet | 95.8 | 79.5 | 36.5M |
| ResNeXt-29 | 96.4 | 82.7 | 68.1M |
| DenseNet-BC | 96.5 | **82.8** | 25.6M |
| **R2Net-N** ($C_4$) | 95.8 | 80.6 | 0.9M |
| **R2Net-S** ($C_4$) | 96.6 | 82.2 | 2.8M |
| **R2Net-M** ($C_4$) | **97.3** | 82.7 | 6.0M |

## 4.2 ADDITIONAL EXPERIMENTS

**Image classification.** Classification evaluation assesses the discriminative quality of features. We construct the backbone R2Net incorporating v8Classify Head for image classification tasks, as de-

picted in Figure 3. Table 8 shows the top-1 accuracy of R2Net in three sizes (-N / S / M) on CIFAR-10 / 100 datasets. R2Net and all outstanding baselines trained from scratch. In CIFAR-10, R2Net-M achieves superior accuracy with significantly fewer parameters than DenseNet-BC, WideResNet, and ResNeXt. In CIFAR-100, R2Net-N has only 2.5% of WideResNet's parameters, R2Net-S has 4.1% of ResNeXt-29's parameters, and R2Net-M has 23.4% of DenseNet-BC's parameters, yet they still achieve competitive accuracy. Note that R2Net effectively balances parameters and accuracy. These results demonstrate the effectiveness of R2GConv. Please refer to Appendix 7.6 for details, and more results of R2Net with different groups (e.g., $\mathbf{C}_2$ / $\mathbf{C}_8$).

**Instance segmentation.** Please refer to Appendix 7.6 for detailed results and analysis.

## 4.3 VISUALIZATION

**Visualization of RRE, SRE and NRE.** The visualization of RRE (Ours), SRE, and Non-Rotation-Equivariance (NRE) is shown in Figure 5. We input (a) into R2Det-N (RRE), R2Det-N (SRE), and YOLOv8-N (NRE) to obtain the feature maps (b), (c), and (d), respectively. Observing the blue circles in (b), we notice slight differences but the overall feature maps are relaxed rotation-equivariant, showcasing the characteristic of our RRE. Observing the orange circles in (c), we find that the feature maps are strict rotation-equivariant, presenting SRE. Lastly, observing the red circles in (d), we find almost the characteristic of NRE. More visualizations can be found in Appendix 7.7.

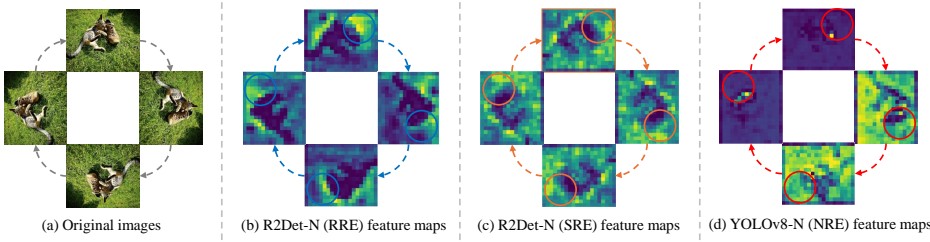

(a) Original images  (b) R2Det-N (RRE) feature maps  (c) R2Det-N (SRE) feature maps  (d) YOLOv8-N (NRE) feature maps

Figure 5: Visualization of the rotated feature maps in RRE (Ours), SRE, and NRE based on $\mathbf{C}_4$.

**Visualization of Rotational Symmetry-Breaking.** We also present a case of Rotational Symmetry-Breaking on $\mathbf{C}_4$ in 2D object detection, as shown in Figure 6. We stretch a tvmonitor with $\mathbf{C}_4$ symmetry, therefore breaking its $\mathbf{C}_4$ symmetry. It can be seen that R2Det with SRE or RRE achieves similar detection probabilities in the unstretched image. However, in the stretched image, the detection probability decreased by 11% for R2Det with SRE, while R2Det with RRE only decreased by 3%. This example indicates that RRE can better model Rotational Symmetry-Breaking situations.

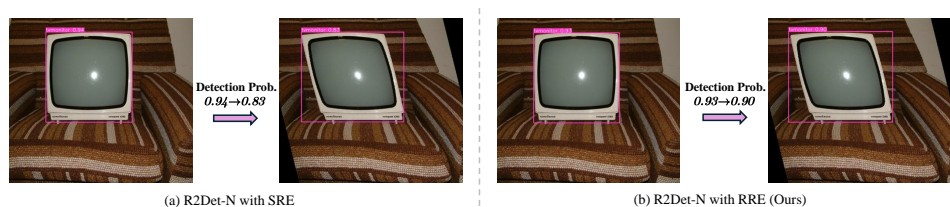

(a) R2Det-N with SRE  (b) R2Det-N with RRE (Ours)

Figure 6: An example of Rotational Symmetry-Breaking on $\mathbf{C}_4$ in 2D object detection.

## 5 CONCLUSION

In this work, we propose a novel approach to build a RRE group $\mathbf{R}_n$ by perturbing a SRE group $\mathbf{C}_n$. Based on $\mathbf{R}_n$, we form a well-designed R2GConv operation, which tackles Rotational Symmetry-Breaking situations to better align with real-world scenarios. Furthermore, we propose an efficient backbone R2Net and a redesigned 2D object detector named R2Det. Experiments demonstrate that our proposed R2Det achieves state-of-the-art performance compared to models without symmetry bias or with SRE constraints in 2D object detection. Additionally, R2Net as a feature extraction network can be extended to more complex visual tasks and scenes, leveraging the advantages of RRE by the proposed R2GConv operation.

## 6 ACKNOWLEDGMENTS

This research is supported by the General Program of Shanghai Natural Science Foundation (No.24ZR1419800, No.23ZR1419300), the National Natural Science Foundation of China (No.42130112, No.42371479), the Science and Technology Commission of Shanghai Municipality (No.22DZ2229004), and Shanghai Frontiers Science Center of Molecule Intelligent Syntheses.

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

## 7 Appendix

### 7.1 Model architecture of R2Det

In this section, we provide details of R2Det-N / S / M, as shown in Table 9.

Table 9: The parameter settings of R2Det-N / S / M. Among them, the **Index** column denotes the module index; the **From** column denotes where the input of this module comes from. For example, $-1$ denotes the upper module, $-1, 6$ denotes the upper module, and 6-module; the **OS** and **OC** columns denote the output size and channels from the upper module, respectively. In the **OC** column, $n$ denotes the group order of our defined group $\mathbf{R}_n$. Note that the channel and size of the first input are 3 and $640 \times 640$, respectively.

| Index | From | Module | OS | R2Det-N | | R2Det-S | | R2Det-M | |
|---|---|---|---|---|---|---|---|---|---|
| | | | | OC | Params. | OC | Params. | OC | Params. |
| 0 | -1 | Lifting R2GConv | $320 \times 320$ | $16n$ | 480 | $32n$ | 944 | 48 | 1408 |
| 1 | -1 | Efficient R2GConv | $160 \times 160$ | $32n$ | 2416 | $64n$ | 8912 | $96n$ | 19504 |
| 2 | -1 | R2Net Block | $160 \times 160$ | $32n$ | 9640 | $64n$ | 37680 | $96n$ | 131152 |
| 3 | -1 | Efficient R2GConv | $80 \times 80$ | $64n$ | 8912 | $128n$ | 34192 | $192n$ | 75856 |
| 4 | -1 | R2Net Block | $80 \times 80$ | $64n$ | 58784 | $128n$ | 232192 | $192n$ | 409664 |
| 5 | -1 | Efficient R2GConv | $40 \times 40$ | $128n$ | 34192 | $256n$ | 133904 | $384n$ | 299152 |
| 6 | -1 | R2Net Block | $40 \times 40$ | $128n$ | 232192 | $256n$ | 923072 | $384n$ | 1630208 |
| 7 | -1 | Efficient R2GConv | $20 \times 20$ | $128n$ | 66960 | $256n$ | 264976 | $384n$ | 594064 |
| 8 | -1 | R2Net Block | $20 \times 20$ | $128n$ | 149056 | $256n$ | 592992 | $384n$ | 2072704 |
| 9 | -1 | R2SPPF | $20 \times 20$ | $128n$ | 65792 | $256n$ | 262656 | $384n$ | 590592 |
| 10 | -1 | Efficient R2GUp | $40 \times 40$ | $64n$ | 33040 | $128n$ | 131600 | $192n$ | 295696 |
| 11 | -1, 6 | Concat | $40 \times 40$ | $192n$ | 0 | $384n$ | 0 | $576n$ | 0 |
| 12 | -1 | R2Net Block | $40 \times 40$ | $128n$ | 181824 | $256n$ | 724064 | $384n$ | 2515072 |
| 13 | -1 | Efficient R2GUp | $80 \times 80$ | $64n$ | 33040 | $128n$ | 131600 | $192n$ | 295696 |
| 14 | -1, 4 | Concat | $80 \times 80$ | $128n$ | 0 | $256n$ | 0 | $384n$ | 0 |
| 15 | -1 | R2Net Block | $80 \times 80$ | $64n$ | 54064 | $128n$ | 214592 | $192n$ | 741472 |
| 16 | -1 | Efficient R2GConv | $40 \times 40$ | $64n$ | 17104 | $128n$ | 66960 | $192n$ | 149584 |
| 17 | -1, 12 | Concat | $40 \times 40$ | $192n$ | 0 | $384n$ | 0 | $576n$ | 0 |
| 18 | -1 | R2Net Block | $40 \times 40$ | $128n$ | 181824 | $256n$ | 724064 | $384n$ | 2515072 |
| 19 | -1 | Efficient R2GConv | $20 \times 20$ | $128n$ | 66960 | $256n$ | 264976 | $384n$ | 594064 |
| 20 | -1, 9 | Concat | $20 \times 20$ | $256n$ | 0 | $512n$ | 0 | $768n$ | 0 |
| 21 | -1 | R2Net Block | $20 \times 20$ | $256n$ | 592992 | $512n$ | 2365600 | $576n$ | 5320864 |
| 22 | 15 | Transfer Block | $80 \times 80$ | $64n$ | 4288 | 128 | 16752 | 192 | 37408 |
| 23 | 18 | Transfer Block | $40 \times 40$ | $128n$ | 16752 | 256 | 66256 | 384 | 148528 |
| 24 | 21 | Transfer Block | $20 \times 20$ | $256n$ | 66256 | 512 | 263568 | 576 | 333376 |
| 25 | 22, 23, 24 | v8Detect Head | - | - | 897664 | - | 2147008 | - | 3822016 |
| | **Total Params** | | | | **2.8M** | | **9.6M** | | **22.6M** |
| | **FLOPs** | | | | **4.0G** | | **9.0G** | | **17.3G** |

### 7.2 Detailed Architecture of R2Det and R2Net.

R2Det leverages the outputs $\{\mathcal{O}_2, \mathcal{O}_3, \mathcal{O}_4\}$ from the last three stages of R2Net as inputs to fuse features across various scales and semantic levels, thereby allowing R2Det to detect objects of diverse sizes within an image. The output $\mathcal{O}_4$ is first refined by an R2SPPF designed for multi-scale spatial max pooling. Then $\mathcal{O}_2$, $\mathcal{O}_3$ and the refined $\mathcal{O}_4$ are fed into a standard FPN+PAN neck part, which includes R2Net Block, Efficient R2GConv, and Efficient R2GUp for $2\times$ upsampling.

Efficient R2GUp adopts the same architecture as Efficient R2GConv but uses the transposed convolution operation for upsampling during the depth-wise convolution stage. R2SPPF, similar to the architecture of R2Net Block, optimizes for parameter efficiency and enhances feature representation by capturing information at various scales.

The obtained features $\{\widetilde{\mathcal{O}}_4, \widetilde{\mathcal{O}}_3, \widetilde{\mathcal{O}}_2\}$ are then input into Transfer Block for channel reduction, preparing for final detection by a universal v8Detector Head of YOLOv8. Transfer Block, which incorporates an Efficient R2GConv, ensures the features are appropriately shaped for the subsequent heads.

v8Detector Head is an anchor-free approach for object's Bounding box (**Bbox**) and Classification (**Cls**) predictions, simplifying the prediction process.

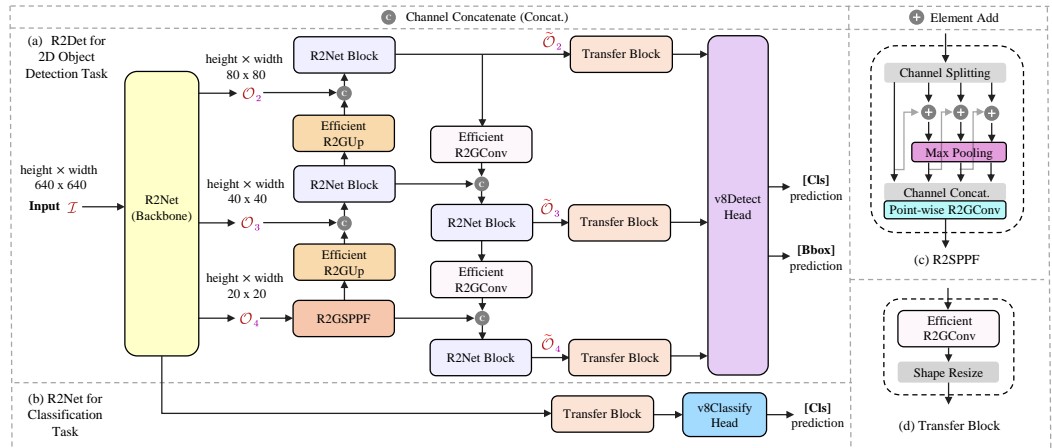

Figure 7: Detailed R2Det for 2D object detection task and R2Net for classification task.

## 7.3 HYPER-PARAMETER SETTINGS

The hyper-parameter settings of R2Det are shown in Table 10. R2Det employs the same hyper-parameter settings of YOLOv8 but only is trained in 300 epochs, which are less than YOLOv8's 500 epochs. And the hyper-parameter settings of R2Net are shown in Table 11.

Table 10: Hyper-parameter settings of R2Det.

| Hyper parameter | Value |
| --- | --- |
| Epochs | 300 |
| Optimizer | SGD |
| Initial learning rate | 0.01 |
| Finish learning rate | 0.0001 |
| Learning rate decay | linear |
| Momentum | 0.937 |
| Weight decay | 0.0005 |
| Warm-up epochs | 3 |
| Warm-up momentum | 0.8 |
| Warm-up bias learning rate | 0.1 |
| Box loss gain | 7.5 |
| Class loss gain | 7.5 |
| DFL loss gain | 1.5 |
| HSV-Hue augmentation | 0.015 |
| HSV-Saturation augmentation | 0.7 |
| HSV-Value augmentation | 0.4 |
| Translation augmentation | 0.1 |
| Scale augmentation | 0.5 |
| Flip left-right augmentation | 0.5 |
| Mosaic augmentation | 1.0 |
| Close mosaic epochs | 10 |

Table 11: Hyper-parameter settings of R2Net.

| Hyper parameter | Value |
| --- | --- |
| Epochs | 200 |
| Optimizer | SGD |
| Initial learning rate | 0.01 |
| Finish learning rate | 0.0001 |
| Learning rate decay | linear |
| Momentum | 0.937 |
| Weight decay | 0.0005 |
| Warm-up epochs | 3 |
| Warm-up momentum | 0.8 |
| Warm-up bias learning rate | 0.1 |
| Class loss gain | 7.5 |
| HSV-Hue augmentation | 0.015 |
| HSV-Saturation augmentation | 0.7 |
| HSV-Value augmentation | 0.4 |
| Translation augmentation | 0.1 |
| Scale augmentation | 0.5 |
| Flip left-right augmentation | 0.5 |
| Mosaic augmentation | 1.0 |
| Close mosaic epochs | 10 |

## 7.4 ANALYSIS ON THE LEARNABLE PERTURBATION PARAMETER $\Delta$

The tendency of $\Delta$ is intrinsically linked to the degree of Rotational Symmetry-Breaking within the system (or dataset). Here we further present the final $\Delta$ values, mean value, and variance from the first four layers of the full-trained Efficient R2GConv in the R2Det-N model in Table 12. Our R2Det

has effectively integrated a Rotational Symmetry-Breaking prior, with $\Delta$ in the proposed R2GConv capable of being updated end-to-end. These $\Delta$ values more accurately reflect Rotational Symmetry-Breaking phenomena in natural datasets, i.e., a minor Relaxed Rotaion-Equivariance (`RRE`).

Table 12: The distribution of $\Delta$ values for four Efficient R2GConv in the full-trained R2Det-N on VOC training dataset. All of them are initialized from the Uniform distribution $\mathcal{U}(-0.1, 0.1)$.

| No | $\Delta_1$ | $\Delta_2$ | $\Delta_3$ | $\Delta_4$ | Mean value | Variance |
|----|-----------|-----------|-----------|-----------|-----------|----------|
| #1 | $\begin{bmatrix} 0.0265 & -0.0041 \\ -0.0127 & 0.0325 \end{bmatrix}$ | $\begin{bmatrix} 0.0310 & -0.0377 \\ 0.0284 & 0.0163 \end{bmatrix}$ | $\begin{bmatrix} -0.0255 & 0.0114 \\ 0.0057 & -0.0343 \end{bmatrix}$ | $\begin{bmatrix} 0.0072 & 0.0317 \\ -0.0310 & -0.0070 \end{bmatrix}$ | 0.0024 | 0.000624 |
| #2 | $\begin{bmatrix} -0.0063 & -0.1191 \\ -0.1504 & -0.1154 \end{bmatrix}$ | $\begin{bmatrix} 0.1082 & 0.0729 \\ -0.0028 & -0.1312 \end{bmatrix}$ | $\begin{bmatrix} 0.0629 & -0.0916 \\ -0.1335 & 0.0094 \end{bmatrix}$ | $\begin{bmatrix} -0.1458 & 0.0432 \\ 0.0275 & 0.0356 \end{bmatrix}$ | -0.0335 | 0.00812 |
| #3 | $\begin{bmatrix} -0.8716 & -0.0249 \\ 0.2800 & -0.0681 \end{bmatrix}$ | $\begin{bmatrix} -0.4702 & 0.0441 \\ -0.2178 & 0.5107 \end{bmatrix}$ | $\begin{bmatrix} 0.0856 & -0.5176 \\ -0.2120 & 0.1603 \end{bmatrix}$ | $\begin{bmatrix} -0.5718 & -0.1320 \\ -0.1283 & -0.0665 \end{bmatrix}$ | -0.138 | 0.119 |
| #4 | $\begin{bmatrix} -0.26733 & 0.32544 \\ 0.59473 & -0.4436 \end{bmatrix}$ | $\begin{bmatrix} -0.21484 & -0.000070453 \\ -0.0061417 & -0.30835 \end{bmatrix}$ | $\begin{bmatrix} 0.033966 & 0.44604 \\ 0.45483 & 0.42603 \end{bmatrix}$ | $\begin{bmatrix} 0.45264 & -0.042358 \\ -0.15881 & 0.18347 \end{bmatrix}$ | 0.0922 | 0.106 |

## 7.5 ANALYSIS ON YOLOV8-N-CLS AND R2NET-N ON ROT-MINIST DATASET.

This section compares the training accuracy of YOLOv8-N-CLS and our R2Net-N on the ROT-MINIST dataset, as shown in the following Table. Note that our designed ROT-MINIST is different from standard Rotated MINST, which aims to test the robustness of our R2Net . Specifically, we manipulate the training set by randomly rotating $60,000$ images by 0, 90, 180, and 270 degrees while maintaining $10,000$ images unaltered in the test set to evaluate the performance of a model under rotation. In Figure 8, both R2Net-N and YOLOv8-N-CLS display fluctuations during training. However, R2Net-N exhibits milder fluctuations compared to the more pronounced oscillations observed in YOLOv8-N-CLS. This contrast highlights the superior rotation anti-interference capability of R2Net-N, which is primarily attributed to its novel Relaxed Rotation-Equivariance (`RRE`).

Table 13: Comparaison of the robustness of YOLOv8-N-CLS and R2Net-N ($\mathbf{C}_4$).

| Method | Dataset | Error | Params. |
|--------|---------|-------|---------|
| YOLOv8-N-CLS | MINST | 0.58 | 1.5M |
| | ROT-MINIST | 44.88 | |
| R2Net-N ($\mathbf{C}_4$) | MINST | **0.54** | 0.8M |
| | ROT-MINIST | **27.75** | |

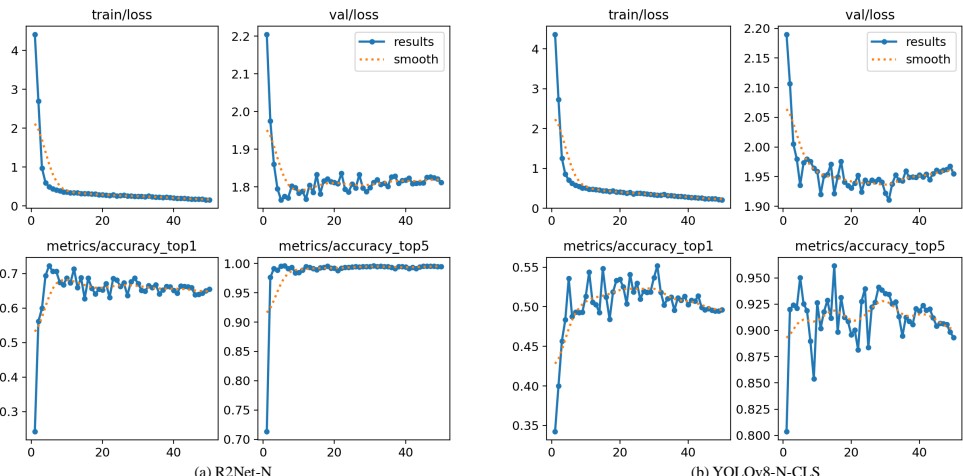

Figure 8: Comparison of YOLOv8-N and our R2Net-N in accuracy on ROT-MINIST dataset. Both models are trained for 50 epochs with the resized input size $224 \times 224$ on dual 4090 GPUs .

## 7.6 ADDITIONAL EXPERIMENTS ON CLASSIFICATION AND INSTANCE SEGMENTATION

**Classification.** Also, we construct the backbone R2Net in three sizes (i.e., -N / S / M) on three groups (i.e., $C_2$ / $C_4$ / $C_8$), incorporating the v8Classify Head for image classification, as shown in Figure 3. Table 8 shows the classification performance of our R2Net on CIFAR-10 / 100 datasets.

R2Net-M ($C_2$) and R2Net-S ($C_4$) both outperform other models in the accuracy of the CIFAR-10 dataset, but their parameters are much fewer. Also, R2Net-S ($C_8$) and R2Net-M ($C_8$) both exceed other models in accuracy of CIFAR-100 dataset. Still, they have fewer parameters compared to other models. We found that R2Net-S ($C_4$) balances the parameter-accuracy trade-offs. These experiments show that R2Net still achieves excellent performance in natural classification tasks.

Table 14: Comparison of other models in top-1 accuracy (%) on CIFAR-10 / 100 datasets.

| Method | CIFAR-10 (%) | CIFAR-100 (%) | Params. |
|---|---|---|---|
| WideResNet | 95.8 | 79.5 | 36.5M |
| ResNeXt-29 | 96.4 | 82.7 | 68.1M |
| DenseNet-BC | 96.5 | 82.8 | 25.6M |
| **R2Net-N ($C_2$)** | 94.3 | 77.4 | 0.6M |
| **R2Net-S ($C_2$)** | 95.9 | 79.5 | 1.7M |
| **R2Net-M ($C_2$)** | 96.7 | 80.5 | 3.4M |
| **R2Net-N ($C_4$)** | 95.8 | 80.6 | 0.9M |
| **R2Net-S ($C_4$)** | 96.6 | 82.2 | 2.8M |
| **R2Net-M ($C_4$)** | 97.3 | 82.7 | 6.0M |
| **R2Net-N ($C_8$)** | 96.2 | 81.5 | 1.4M |
| **R2Net-S ($C_8$)** | 96.9 | 83.6 | 5.0M |
| **R2Net-M ($C_8$)** | **97.7** | **84.4** | 11.2M |

Table 15: Comparison of other models in instance segmentation on COCO-seg dataset.

| Method | Box AP(%) | Box AP$_{50}$(%) | Mask AP(%) | Mask AP$_{50}$(%) | Params. | FLOPs |
|---|---|---|---|---|---|---|
| YOLOv8-N-seg | 36.2 | 51.2 | 29.6 | 48.2 | 3.4M | 12.6G |
| YOLOv8-S-seg | 44.0 | 60.4 | 36.0 | 56.8 | 11.8M | 42.6G |
| RTMDet-Ins-Tiny | 40.5 | - | 35.4 | - | 5.6M | 11.8G |
| RTMDet-Ins-S | 44.0 | - | 38.7 | - | 10.2M | 21.5G |
| R2Det-N-seg ($C_4$) | 43.7 | 59.3 | 35.8 | 56.2 | 3.0M | 7.9G |

● Note that R2Det-N-seg adopts the same architecture of R2Det-N but replaces *v8Detect Head* with *v8Segment Head*.

**Instance Segmentation.** We also conduct instance segmentation tasks on the COCO-seg dataset, as shown in Table 15. Compared to YOLOv8-N-seg, our R2Det-N-seg exhibits improvements of 20.7% in **Box AP** and 20.9% in **Mask AP**, with fewer parameters. R2Det-N-seg achieves similar **Mask AP** of YOLOv8-S-seg and RTMDet-Ins-Tiny, with only 25.4% and 53.6% of their parameters. Moreover, when compared to RTMDet-Ins-S, R2Det-N-seg has similar **Box AP** while utilizing only 29.4% parameters. The results show that our R2Det is also applicable to instance segmentation.

## 7.7 VISUALIZATION ANALYSIS ON RELAXED ROTATION-EQUIVARIANCE

In this subsection, we present a visualization of feature maps from our R2Det-N, as illustrated in Figure 9. We rotate the initial image (a) by 90, 180, and 270 degrees to generate images (b), (c), and (d) as inputs. It can be found that the output feature maps in (e), (f), (g), and (h), corresponding to each channel, exhibit consistency with minor variations, which demonstrates the Relaxed Rotation-Equivariance (RRE) property of our network.

## 7.8 HEATMAP VISUALIZATION

In this section, we present the visualization of LayerCAM (Jiang et al., 2021) heatmaps derived from YOLOv8-N, YOLOv7, YOLOv5, and our R2Det-N ($C_4$), as depicted in Figure 10. These heatmaps enable us to locate the regions of interest where the network concentrates its attention. It can be seen that YOLOv7 and our R2Det-N achieved better feature focusing. Notably, R2Det-N shows a comprehensive focusing range on certain objects, such as dogs and zebras. In contrast, YOLOv8 and YOLOv5 fail to exhibit such targeted feature focus on these particular objects.

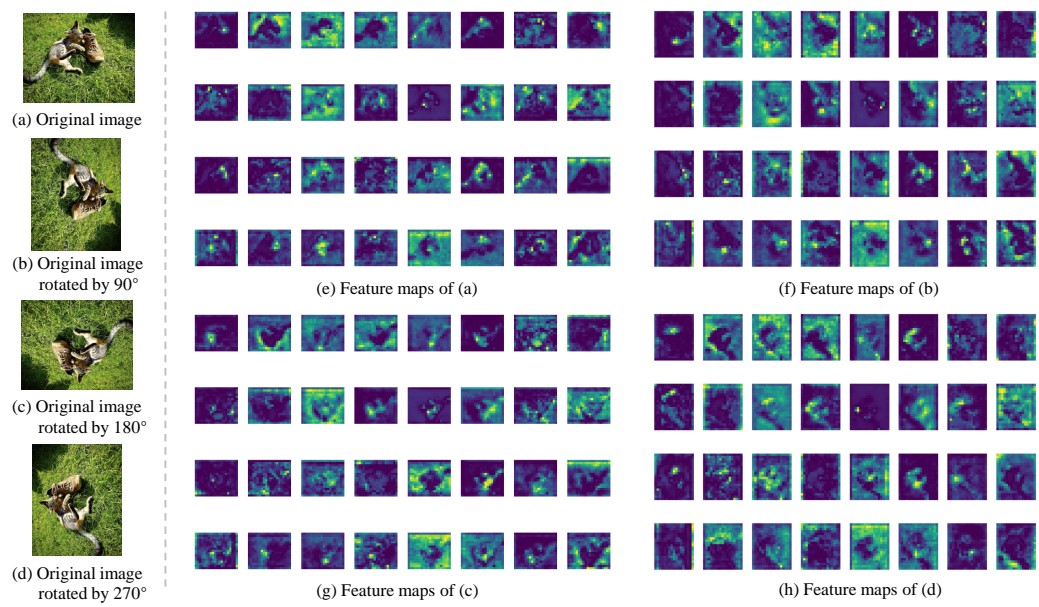

Figure 9: The R2Det-N ($\mathbf{C}_4$) feature map visualization of the original image rotated at (a) 0, (b) 90, (c) 180, and (d) 270 degrees, as depicted in (e), (f), (g), and (h), corresponds to its 32 channels.

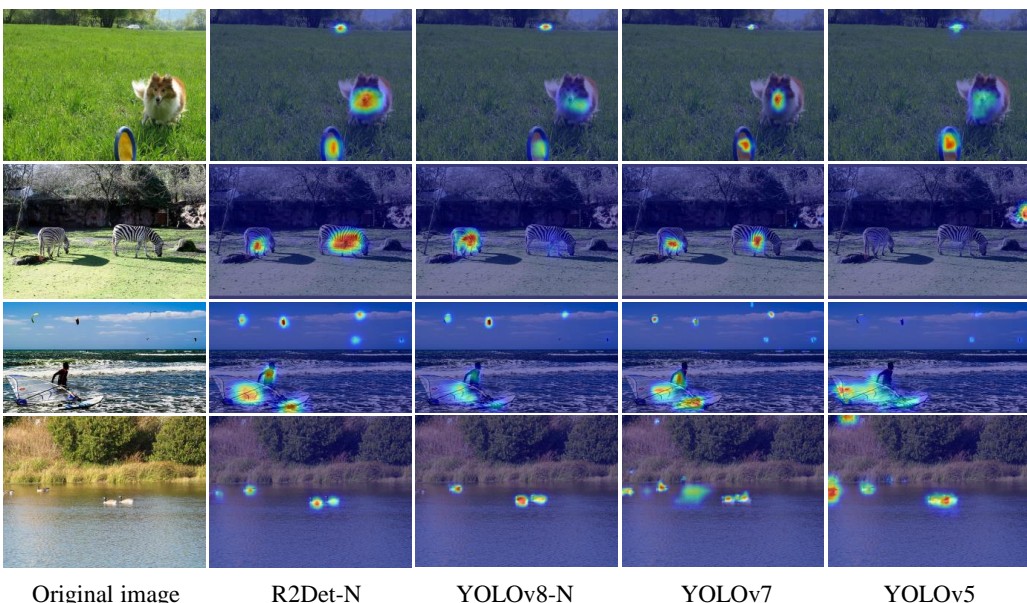

Figure 10: LayerCAM heatmaps visualizations of neck networks across YOLO (v8-N, v7, v5) and our R2Det-N ($\mathbf{C}_4$) models. Examples are from the COCO dataset.

## 7.9 PARAMETER ANALYSIS OF EFFICIENT R2GCONV (OURS) AND GCONV ON $\mathbf{C}_n$

Assuming the input, output channels, and kernel size of both Efficient R2GConv and GConv are $c_{in}$, $c_{out}$, and $k$, respectively. The parameters of Efficient R2GConv can be calculated as follows:

$$c_{in} \times c_{out} \times n \times 1 \times 1 \,(\textbf{Point-wise R2GConv}) + c_{out} \times 1 \times 1 \times k \times k \,(\textbf{Depth-wise R2GConv})$$
$$+ n \times 2 \times 2 \,(\Delta) \approx n \times c_{in} \times c_{out} + k^2 \times c_{out}$$

where $\cdot$ in $(\cdot)$ denotes the source of parameters.

The parameters of GConv can be calculated as follows:

$$c_{in} \times c_{out} \times n \times k \times k = n \times c_{in} \times c_{out} \times k^2.$$

Therefore, the parameter of our Efficient R2GConv is only

$$\frac{n \times c_{in} \times c_{out} + k^2 \times c_{out}}{n \times c_{in} \times c_{out} \times k^2} = \frac{1}{k^2} + \frac{1}{n \times c_{in}}$$

of GConv.

## 7.10 PROOF OF GCONV (SRE) AND R2GCONV (RRE)

**Definition 3 (*Approximate Equivariance*).** (Wang et al., 2022a) *A learning function $\phi_{approx} : X \to Y$ that sends elements from input space $X$ to output space $Y$ satisfies Approximate Equivariance to a group $G$ if $\forall g, \mathbf{x} \in G \times X$ there exists $\rho_X : G \to \mathrm{GL}(X)$ and $\rho_Y : G \to \mathrm{GL}(Y)$ actions of $G$ such that*

$$\|\phi_{approx}(\rho_X(g) \cdot \mathbf{x}) - \rho_Y(g) \cdot \phi_{approx}(\mathbf{x})\| \le \epsilon, \tag{9}$$

*where a small $\epsilon$ indicates strong equivariance and a relatively larger (not very large) $\epsilon$ exhibits greater relaxation. Especially, $\phi_{approx}$ is equivalent to $\phi_{strict}$ and also satisfies Eq. 1 when $\epsilon = 0$.*

In effect, Approximate Equivariance can be seen as Relaxed Equivariance since they are defined by different perspectives. Approximate Equivariance is defined from the L2 Norm, which can be computable. Relaxed Equivariance is defined from the subgroup, which is universally applicable but more abstract. In this paper, we use the definition of Approximate Equivariance to prove the following conclusions.

On $\mathbf{C}_n$ group, define $\mathbf{c}^i(\cdot)$ as rotation of $\cdot$ by $2\pi i/n$, and $\mathbf{c}^{i+1}(\cdot) = \mathbf{c}^{(i+1) \bmod n}(\cdot)$. We have the following conclusion (note that for simplicity, we ignore the input and output channels):

- **Conclusion 1.** $\mathbf{C}_n$-GConv (Vanilla Group Convolution) is a strict rotation-equivariant block.

   **Proof.** Given input $x$ and initial weight $\psi$, we obtain the strict rotation-equivariant filter $\psi_i^{\text{strict}} = \mathbf{c}^i(\psi)$ in the $i$-order of $\mathbf{C}_n$. Then $\mathbf{C}_n$-GConv can be defined as: $f_1(x) = \sum_{i=0}^{n-1} x * \psi_i^{\text{strict}} = \sum_{i=0}^{n-1} x * \mathbf{c}^i(\psi)$. For any $j \in \{0, 1, \cdots, n-1\}$, we have

   - $f_1(\mathbf{c}^j(x)) = \sum_{i=0}^{n-1} \mathbf{c}^j(x) * \mathbf{c}^i(\psi)$
   - $\mathbf{c}^j(f_1(x)) = \mathbf{c}^j(\sum_{i=0}^{n-1} x * \mathbf{c}^i(\psi)) = \sum_{i=0}^{n-1} \mathbf{c}^j(x) * \mathbf{c}^{i+j}(\psi)$
     $= \sum_{i=0}^{n-1} \mathbf{c}^j(x) * \mathbf{c}^{(i+j) \bmod n}(\psi) = \sum_{i=0}^{n-1} \mathbf{c}^j(x) * \mathbf{c}^i(\psi).$

   Therefore, $f_1(\mathbf{c}^j(x)) = \mathbf{c}^j(f_1(x))$. According to Eq. 1, complete the proof.

- **Conclusion 2.** $\mathbf{C}_n$-R2GConv (Ours) is a relaxed rotation-equivariant block.

   **Proof.** Given input $x$, initial weight $\psi$, an affine transformation function $\mathbf{t}$ and the learnable perturbation $\Delta$, we obtain the relaxed rotation-equivariant filter $\psi_i^{\text{relaxed}} = \mathbf{t}^i(\psi, \Delta)$ in the $i$-order of $\mathbf{C}_n$. Then $\mathbf{C}_n$-R2GConv can be defined as: $f_2(x) = \sum_{i=0}^{n-1} x * \psi_i^{\text{relaxed}} = \sum_{i=0}^{n-1} x * \mathbf{t}^i(\psi, \Delta)$. For any $j \in \{0, 1, \cdots, n-1\}$, we have:

   - $f_2(\mathbf{c}^j(x)) = \sum_{i=0}^{n-1} \mathbf{c}^j(x) * \mathbf{t}^i(\psi, \Delta)$
   - $\mathbf{c}^j(f_2(x)) = \mathbf{c}^j(\sum_{i=0}^{n-1} x * \mathbf{t}^i(\psi, \Delta)) = \sum_{i=0}^{n-1} \mathbf{c}^j(x) * \mathbf{c}^j(\mathbf{t}^i(\psi, \Delta)).$

   Therefore, $\|f_2(\mathbf{c}^j(x)) - \mathbf{c}^j(f_2(x))\| = \|\sum_{i=0}^{n-1} \mathbf{c}^j(x) * \mathbf{t}^i(\psi, \Delta) - \sum_{i=0}^{n-1} \mathbf{c}^j(x) * \mathbf{c}^j(\mathbf{t}^i(\psi, \Delta))\| = \|\sum_{i=0}^{n-1} \mathbf{c}^j(x) * (\mathbf{t}^i(\psi, \Delta) - \mathbf{c}^j(\mathbf{t}^i(\psi, \Delta)))\| \le \epsilon$. According to Eq. **??**, complete the proof.
   In particular, when $\Delta = 0$, we have $\mathbf{t}^i(\psi, \Delta) = \mathbf{c}^i(\psi)$, thus $\|f_2(\mathbf{c}^j(x)) - \mathbf{c}^j(f_2(x))\| = 0$, i.e., $\epsilon = 0$, and $f_2$ is strict rotation-equivariant when $\Delta = 0$.

## 7.11 THEORETICAL ANALYSIS

Since the existing methods, including strict rotation-equivariant models, cannot perfectly tackle Symmetry-Breaking scenarios in object detection tasks, i.e., they can not learn a relaxed equivariant function. We assume from Wang et al. (2022a) that the ground truth function, named $\phi_{gt}$, is

relaxed equivariant. Firstly, we provide the Equivariance Error (EE), which quantifies how much the ground truth equivariant function $\phi_{gt}$ deviates from being strictly equivariant. The EE is defined as the maximum deviation from the strict-equivariant behavior under transformations, thereby illustrating how we can regulate the level of relaxation in the equivariance property.

**Definition 7.1 (*Equivariance Error*)** *Let $\phi_{gt}\colon X \to Y$ be a function and $G$ be a group. Assume that $G$ acts on $X$ and $Y$ via $\rho_X\colon G \to \mathrm{GL}(X)$ and $\rho_Y\colon G \to \mathrm{GL}(Y)$. For any $g, \mathbf{x} \in G \times X$, the Equivariance Error of $\phi_{gt}$ is defined as follow:*

$$\|\phi_{gt}\|_{\mathrm{EE}} = \sup_{\mathbf{x},g} \|\rho_Y(g) \cdot \phi_{gt}(\mathbf{x}) - \phi_{gt}(\rho_X(g) \cdot \mathbf{x})\|,$$

where $\|\cdot\|$ denote the `L2-Norm` operation. According to Equation **??** in the main text, the function $\phi_{gt}$ is relaxed (or $\epsilon$-approximate) equivariant if and only if $\|\phi_{gt}\|_{\mathrm{EE}} \leq \epsilon$, where $\epsilon$ represents the maximum level of deviation from strict equivariance for $\phi_{gt}$ to be considered relaxed equivariant.

Next, we prove that if an input is close to its transformed version, the images under a continuous relaxed equivariant function still have to be close. This captures the idea of symmetry in the task, meaning that the output of the function is predictable under transformations of the input.

**Proposition 7.2** *Let $\phi_{gt}$ be relaxed (or $\epsilon$-approximate) equivariant and Lipschitz with constant $k$. Then, we have*

$$\|\rho_Y(g) \cdot \phi_{gt}(\mathbf{x}) - \phi_{gt}(\mathbf{x})\| \leq k\|\rho_X(g) \cdot \mathbf{x} - \mathbf{x}\| + \epsilon, \quad \forall g, \mathbf{x} \in G \times X.$$

**Proof.** If $\phi_{gt}$ is Lipschitz with constant $k$, we have

$$\|\phi_{gt}(\rho_X(g) \cdot \mathbf{x}) - \phi_{gt}(\mathbf{x})\| \leq k\|\rho_X(g) \cdot \mathbf{x} - \mathbf{x}\|, \quad \forall g, \mathbf{x} \in G \times X.$$

Further, from the EE definition and triangle inequality, we have

$$\begin{aligned}
\|\rho_Y(g) \cdot \phi_{gt}(\mathbf{x}) - \phi_{gt}(\mathbf{x})\| &\leq \|\rho_Y(g) \cdot \phi_{gt}(\mathbf{x}) - \phi_{gt}(\rho_X(g) \cdot \mathbf{x})\| \\
&\quad + \|\phi_{gt}(\rho_X(g) \cdot \mathbf{x}) - \phi_{gt}(\mathbf{x})\| \\
&\leq k\|\rho_X(g) \cdot \mathbf{x} - \mathbf{x}\| + \epsilon, \quad \forall g, \mathbf{x} \in G \times X.
\end{aligned}$$

$\square$

Note that both terms, $k\|\rho_X(g) \cdot \mathbf{x} - \mathbf{x}\|$ and $\epsilon$, collectively determine the overall upper limit of the equivariance error. The former term embodies the fundamental discrepancy inherently introduced by the specific transformation actions, which naturally exist in the real world and are random. Hence, in this work, we assume $\rho$ as learnable permutations that can be modeled as variables $\Delta$ following a Uniform distribution, i.e., $\Delta \sim \mathcal{U}(-b, b)$. These learnable transformations are also inherently norm-conserving, thereby allowing us to incorporate $\rho$ into our considerations implicitly. Consequently, when an input $\mathbf{x}$ exhibits proximity to its transformed version, the outputs under a continuously equivariant function will also maintain close. We further provide visualization experiments in Seciton 4.3 and Appendix 7.7 as evidence. Moreover, $\phi_{gt}$ does not require maintaining all the symmetries of the input, in contrast to strict equivariance, which imposes constraints on the stabilizer of the output. Finally, since the model $\phi_{relaxed}$ aims to approximates $\phi_{gt}$, the following proposition shows that the equivariance error of the $\phi_{relaxed}$ will converge to the $\|\phi_{gt}\|_{\mathrm{EE}}$.

**Proposition 7.3** *Let $\phi_{gt}\colon X \to Y$ be a function with $\|\phi_{gt}\|_{\mathrm{EE}} = \epsilon$. Assume $\|\phi_{gt} - \phi_{relaxed}\|_{\infty} \leq c$. Then $\|\rho_Y(g) \cdot \phi_{relaxed}(\mathbf{x}) - \phi_{relaxed}(\rho_X(g) \cdot \mathbf{x})\| \leq 2c + \epsilon$.*

**Proof.** By triangle inequality and invariance of the `L2-Norm`, we have

$$\begin{aligned}
\|\rho_Y(g) \cdot \phi_{relaxed}(\mathbf{x}) - \phi_{relaxed}(\rho_X(g) \cdot \mathbf{x})\| &\leq \|\rho_Y(g) \cdot \phi_{relaxed}(\mathbf{x}) - \rho_Y(g) \cdot \phi_{gt}(\mathbf{x})\| \\
&\quad + \|\rho_Y(g) \cdot \phi_{gt}(\mathbf{x}) - \phi_{gt}(\rho_X(g) \cdot \mathbf{x})\| \\
&\quad + \|\phi_{gt}(\rho_X(g) \cdot \mathbf{x}) - \phi_{relaxed}(\rho_X(g) \cdot \mathbf{x})\| \\
&\leq c + \epsilon + c = 2c + \epsilon, \quad \forall g, \mathbf{x} \in G \times X.
\end{aligned}$$

$\square$

## 7.12 MORE COMPARISON OF STATE-OF-THE-ART OBJECT DETECTORS.

We also provide a more detailed comparison of state-of-the-art object detectors, e.g., YOLOv5, YOLOv6, YOLOv7, and Gold YOLO models, as shown in Table 16.

Table 16: Comparison of state-of-the-art object detectors on the COCO validation dataset.

| Method | $\text{AP}_{50:95}(\%)$ | $\text{AP}_{50}(\%)$ | $\text{AP}_{75}(\%)$ | $\text{AP}_{S}(\%)$ | $\text{AP}_{M}(\%)$ | $\text{AP}_{L}(\%)$ | Params. | FLOPs |
|---|---|---|---|---|---|---|---|---|
| YOLOv5u-N | 34.3 | 49.7 | 37.2 | 16.8 | 38.1 | 48.4 | 2.6M | 7.7G |
| YOLOv5u-S | 43.1 | 59.9 | 47.2 | 24.7 | 47.6 | 58.4 | 9.1M | 24.0G |
| YOLOv5u-M | 49.1 | 66.0 | 53.8 | 31.2 | 54.2 | 65.4 | 25.1M | 64.2G |
| YOLOv5u-L | 52.3 | 69.2 | 57.2 | 34.8 | 57.9 | 69.1 | 53.2M | 135.0G |
| YOLOv5u-X | 53.3 | 70.2 | 58.2 | 36.9 | 58.9 | 69.3 | 97.2M | 246.4G |
| YOLOv6-N v3.0 | 37.0 | 52.7 | – | – | – | – | 4.7M | 11.4G |
| YOLOv6-S v3.0 | 44.3 | 61.2 | – | – | – | – | 18.5M | 45.3G |
| YOLOv6-M v3.0 | 49.1 | 66.1 | – | – | – | – | 34.9M | 85.8G |
| YOLOv6-L v3.0 | 51.8 | 69.2 | – | – | – | – | 59.6M | 150.7G |
| YOLOv7 | 51.2 | 69.7 | 55.9 | 31.8 | 55.5 | 65.0 | 36.9M | 104.7G |
| YOLOv7-X | 52.9 | 71.1 | 51.4 | 36.9 | 57.7 | 68.6 | 71.3M | 189.9G |
| YOLOv7-N AF | 37.6 | 53.3 | 40.6 | 18.7 | 41.7 | 52.8 | 3.1M | 8.7G |
| YOLOv7-S AF | 45.1 | 61.8 | 48.9 | 25.7 | 50.2 | 61.2 | 11.0M | 28.1G |
| YOLOv7 AF | 53.0 | 70.2 | 57.5 | 35.8 | 58.7 | 68.9 | 43.6M | 130.5G |
| YOLOv8-N | 37.3 | 52.6 | 40.5 | 18.6 | 41.0 | 53.5 | 3.2M | 8.7G |
| YOLOv8-S | 44.9 | 61.8 | 48.7 | 26.0 | 49.9 | 61.1 | 11.2M | 28.6G |
| YOLOv8-M | 50.2 | 67.2 | 54.7 | 32.3 | 55.9 | 66.5 | 25.9M | 78.9G |
| YOLOv8-L | 52.9 | 69.8 | 57.5 | 35.3 | 58.3 | 69.8 | 43.7M | 165.2G |
| YOLOv8-X | 53.9 | 71.0 | 58.7 | 35.7 | 59.3 | 70.7 | 68.2M | 257.8G |
| Gold YOLO-N | 39.6 | 55.7 | – | 19.7 | 44.1 | 57.0 | 5.6M | 12.1G |
| Gold YOLO-S | 45.4 | 62.5 | – | 25.3 | 50.2 | 62.6 | 21.5M | 46.0G |
| Gold YOLO-M | 49.8 | 67.0 | – | 32.3 | 55.3 | 66.3 | 41.3M | 87.5G |
| Gold YOLO-L | 51.8 | 68.9 | – | 34.1 | 57.4 | 68.2 | 75.1M | 151.7G |
| YOLO MS-N | 43.4 | 60.4 | 47.6 | 23.7 | 48.3 | 60.3 | 4.5M | 17.4G |
| YOLO MS-S | 46.2 | 63.7 | 50.5 | 26.9 | 50.5 | 63.0 | 8.1M | 31.2G |
| YOLO MS | 51.0 | 68.6 | 55.7 | 33.1 | 56.1 | 66.5 | 22.2M | 80.2G |
| GELAN-S | 46.7 | 63.0 | 50.7 | 25.9 | 51.5 | 64.0 | 7.1M | 26.4G |
| GELAN-M | 51.1 | 67.9 | 55.7 | 33.6 | 56.4 | 67.3 | 20.0M | 76.3G |
| GELAN-C | 52.5 | 69.5 | 57.3 | 35.8 | 57.6 | 69.4 | 25.3M | 102.1G |
| GELAN-E | 55.0 | 71.9 | 60.0 | 38.0 | 60.6 | 70.9 | 57.3M | 189.0G |
| YOLOv9-S | 46.8 | 63.4 | 50.7 | 26.6 | 56.0 | 64.5 | 7.1M | 26.4G |
| YOLOv9-M | 51.4 | 68.1 | 56.1 | 33.6 | 57.0 | 68.0 | 20.0M | 76.3G |
| YOLOv9-C | 53.0 | 70.2 | 57.8 | 36.2 | 58.5 | 69.3 | 25.3M | 102.1G |
| YOLOv9-E | 55.6 | 72.8 | 60.6 | 40.2 | 61.0 | 71.4 | 57.3M | 189.0G |
| YOLOv10-N | 38.5 | 53.8 | 41.7 | 18.9 | 42.4 | 54.6 | 2.3M | 6.7G |
| YOLOv10-S | 46.3 | 63.0 | 50.4 | 26.9 | 51.1 | 63.7 | 7.2M | 21.6G |
| YOLOv10-M | 51.1 | 68.1 | 55.8 | 33.8 | 56.5 | 67.0 | 15.4M | 59.1G |
| YOLOv10-B | 52.5 | 69.6 | 57.2 | 35.1 | 57.8 | 68.4 | 19.1M | 92.0G |
| YOLOv10-L | 53.2 | 70.1 | 58.0 | 35.7 | 58.4 | 69.4 | 24.4M | 120.3G |
| YOLOv10-X | 54.4 | 71.4 | 59.4 | 37.1 | 59.9 | 71.1 | 29.5M | 160.4G |
| $\text{C}_4$-R2Det-N | 43.7 | 59.5 | 47.8 | 24.2 | 48.2 | 59.3 | 2.8M | 4.0G |
| $\text{C}_4$-R2Det-S | 50.0 | 66.5 | 54.6 | 30.5 | 55.7 | 66.2 | 9.6M | 9.0G |
| $\text{C}_4$-R2Det-M | 53.1 | 70.3 | 57.9 | 36.4 | 58.7 | 69.6 | 22.6M | 17.3G |

