# OpenReview forum: "R2Det: Exploring Relaxed Rotation Equivariance in 2D Object Detection"
_ICLR.cc/2025/Conference — ICLR 2025 Poster_

### Official Review · Reviewer_AppE · 2024-10-30

**Soundness:** 4
**Presentation:** 3
**Contribution:** 3
**Rating:** 8
**Confidence:** 3

**Summary:**

The paper first discusses a limitation of strict equivariance for real-world applications (i.e. symmetry breaking) and then introduces building blocks and architectures for relaxed rotation equivariance. It focuses mostly on 2D object detection.
More specifically, the authors introduce a relaxation of the group operations on the $n$-order cyclic rotation group $C_n$ by adding learnable perturbations to existing (strict) rotation equivariant filters. They then design different relaxed rotation-equivariant group convolutional modules that serve as building blocks for two architectures: the relaxed rotation-equivariant network (R2Net) and object detector (R2Det).

Experimentally, the paper compares different perturbation levels and shows that a small perturbation leads to stronger performance than strict equivariance (i.e., with no perturbation) on 2D detection datasets (PASCAL VOC and COCO). The new method is also compared to the YOLO series detectors and achieves much higher performance-compute trade-offs. Finally, the R2Net and R2Det methods are evaluated respectively on image classification and instance segmentation.

**Strengths:**

- The method demonstrates impressive performance-compute trade-offs on the object detection task.
- The authors evaluated the effect of the main contribution (i.e., the learnable perturbation) on performance. Furthermore, its effect on the learned features is well illustrated in Figure 5.

**Weaknesses:**

1. The provided background and naming of methods are misleading and partially incorrect. Indeed, **relaxed** equivariance is defined in (Kaba and Ravanbakhsh, 2023) as a relaxation that allows breaking the symmetry of inputs and mapping to arbitrary orbit types when necessary. Note that the output of the function is still predictable under the transformation of the input. While in R2Det, the relaxed equivariance is mistakenly defined (definition 1, line 161) using the definition of **$\epsilon$-approximate** equivariance (despite referring to the definition from (Wang et al., 2022a) which correctly names it **$\epsilon$-approximate** equivariance).
Furthermore, the introduced filter is called a "relaxed rotation-equivariant filter" but is implemented by allowing for some learnable perturbation, which is therefore **NOT** a **relaxed** rotation-equivariance module. Figure 1a also incorrectly illustrates the problem being tackled in the paper. Notably, relaxed equivariance was introduced as an alternative to noise-injection methods: "offering an alternative to the noise-injection methods" (see the abstract from Kaba and Ravanbakhsh, 2023).
The above-mentioned problems make the paper's claims incorrect and could lead to important misunderstandings of already established concepts.
2. The paper is not easy to read, mainly because of: the abundant use of abbreviations and acronyms (ENN, GConv, RRE, SRE, NRE, R2Filter, R2Lift, R2GConv, DR2GConv, PR2GConv, ER2GConv, ER2GCBA, ...), multiple typos  ("Rotationa-Equivariant" line 25, "we further exploring" line 62, "More analysis can refer to" line 167, "converge when 66-epoch" and  "converges at about 198 in the epoch" lines 401-402, ...) and over-loaded illustrations (e.g., Figure 2 and 3).

**Questions:**

Suggestions:
- (Weaknesses 1.): The reviewer suggests that the authors fix the mistakes in the preliminary section, and rename the methods and claims accordingly.
- (Weaknesses 2.): The reviewer suggests the authors to fix typos, to reduce the number of acronyms in plain text (e.g., define only R2GConv and call its variants "point-wise R2GConv", "efficient R2GConv", ...), and to only include *simple* and *intuitive* figures in the main article to simplify the reading. The more detailed diagrams could then be moved to appendix.

Questions:
- The performance gap between different $n$-norder cyclic rotation groups is substantial (Table 3). What is the reason for such a gap? Is it only due to the "newer equivariant angles"? How would the introduced architecture perform without equivariance? Also, is there an intuitive explanation for why it is beneficial for 2D object detection (with non-rotating bounding boxes, like on COCO) to use $C_8$ instead of $C_4$, despite the output (bounding boxes) only having symmetries in the $C_4$ group?
- Experiments showed that on both PASCAL VOC and COCO, performance with strict equivariance is inferior to that with the approximate equivariance introduced in the paper. For intuition, could the authors provide specific (intuitive or experimental) examples of 2D object detection in which it is beneficial to break the strict rotation equivariance?

I am willing to raise my rating if my concerns are addressed.

**Details Of Ethics Concerns:**

Potential (but unlikely) plagiarism: the submission is very similar to a preprint that has been on arXiv [1] since August 2024, and Figure 8 even mistakenly re-uses the method name from the ArXiv paper. It would be useful to verify that the authors of the ICLR paper and arXiv paper are the same, which would mean that they just changed their own paper title and method name.

[1]: "Wu, Zhiqiang, et al. "SBDet: A Symmetry-Breaking Object Detector via Relaxed Rotation-Equivariance." arXiv preprint arXiv:2408.11760 (2024).".

---

> ### Author Response · Authors · 2024-11-17
> **Author Rebuttal to Reviewer AppE: Part 1**
>
> **Thank you for taking the valuable time to review the manuscript. The following may address your concerns.**
>
> > **Weakness 1:**
> > The provided background and naming of methods are misleading and partially incorrect.
> > - **Weakness 1.1:** Indeed, relaxed equivariance is defined in (Kaba and Ravanbakhsh, 2023) as a relaxation that allows breaking the symmetry of inputs and mapping to arbitrary orbit types when necessary.
> > - **Weakness 1.2:** Note that the output of the function is still predictable under the transformation of the input.
> > - **Weakness 1.3:** While in R2Det, the relaxed equivariance is mistakenly defined (definition 1, line 161) using the definition of approximate equivariance (despite referring to the definition from (Wang et al., 2022a) which correctly names it $\epsilon$-approximate equivariance).
> > - **Weakness 1.4:** Furthermore, the introduced filter is called a "relaxed rotation-equivariant filter" but is implemented by allowing for some learnable perturbation, which is therefore NOT a relaxed rotation-equivariance module.
> > - **Weakness 1.5:** Figure 1a also incorrectly illustrates the problem being tackled in the paper.
> > - **Weakness 1.6:** Notably, relaxed equivariance was introduced as an alternative to noise-injection methods: "offering an alternative to the noise-injection methods" (see the abstract from Kaba and Ravanbakhsh, 2023).
> > - **Weakness 1.7:** The above-mentioned problems make the paper's claims incorrect and could lead to important misunderstandings of already established concepts.
>
> **Answers to Weakness 1.1/1.3/1.5/1.6/1.7:**
> Sorry for the confusion.
>
> In the revised version of our paper, we have explicitly provided the definitions of relaxed equivariance from (Kaba and Ravanbakhsh, 2023) and approximate equivariance from (Wang et al., 2022a). Please refer to the revised version of our paper.
>
> Relaxed equivariance is defined in (Kaba and Ravanbakhsh, 2023), where the concept of noise-injection is mentioned as **a method** to construct relaxed equivariant networks. The noise is added to the data before processing it through an equivariant network.
>
> Relaxed equivariance allows for breaking the symmetry of inputs and mapping to arbitrary orbit types when necessary. The key aspect is that the output of the function remains predictable under the transformation of the input. This definition emphasizes flexibility in handling symmetries while maintaining predictability.
>
> On the other hand, approximate equivariance, as defined by Wang et al. (2022a), focuses on the similarity in the output under the same group transformation. It allows for small deviations in the output, making it more practical for real-world applications.
>
> **In fact, approximate equivariance is a case of relaxed equivariance, and they are actually solving the same problem, i.e., the symmetry-breaking. The definition of relaxation is broader, but it is highly abstract and difficult to measure the degree of relaxation; Wang et al. define approximate equivariance from the computable perspective of L2 norm, using $\epsilon$ to measure the degree of relaxation, which is more applicable.**
>
> Therefore, in this work, we assume that the symmetry-breaking encountered in visual image data under the rotation group, as introduced by Kaba and Ravanbakhsh (2023), also satisfies the definition of approximate equivariance from Wang et al. (2022a). This assumption allows us to introduce and implement the "relaxed rotation-equivariant filter" while retaining the benefits of both relaxed and approximate equivariance.
>
> Based on the above definitions, we proposed a relaxed rotation-equivariant filter. This filter is designed to allow for some learnable perturbation, which is crucial for maintaining the flexibility of the relaxed equivariance framework.
>
> In the latest revised version, we have clarified these issues, including precisely defining relaxed and approximate equivariance and refining any content that may have caused reader misunderstandings. We have also elaborated on how we have leveraged these concepts to propose and implement the relaxed rotation-equivariant filter.

---

> ### Author Response · Authors · 2024-11-17
> **Author Rebuttal to Reviewer AppE: Part 2**
>
> **Answer to Weakness 1.2**:
> Introducing the learnable perturbation $\Delta$ may lead to the misconception that R2Det is unpredictable, but our R2Det is indeed predictable.
> The reason for this misunderstanding might stem from the belief that $\Delta$ is unpredictable.
>
> **In fact, $\Delta$ can be considered as implicitly predictable.**
> **During the training phase, $\Delta$ is updated end-to-end through gradient descent, a process determined by the training data.**
>
> Once the model completes training, gradient updates cease, and the model's parameters are frozen (including $\Delta$), thus making the entire model **a deterministic function.**
>
> Therefore, the output of a deterministic function is predictable under the transformation of the input,
> which is consistent with the discussion of Relaxed Equivariance in (Kaba and Ravanbakhsh, 2023).
>
> **Answer to Weakness 1.4**:
> On $\mathbf{C}_n$ group, define $\mathbf{c}^i(\cdot)$ as rotation of $\cdot$ by $2\pi i/n$, and
> $\mathbf{c}^{i+1}(\cdot)=\mathbf{c}^{(i+1)\~\text{mod}\~n}(\cdot)$.
> We have the following conclusion (Note: for simplicity, we ignore the input and output channels):
>
> **1. $\mathbf{C}_n$-GConv (Vanilla Group Convolution) is a strict rotation-equivariant block, proven as follows:**
>
> Given input $x$ and initial weight $\psi$, we obtain the strict rotation-equivariant filter
> $\psi_i^{\text{strict}}=\mathbf{c}^i(\psi)$ in the $i$-order of $\mathbf{C}\_n$.
> Then $\mathbf{C}\_n$-GConv can be defined as:
> $f_1(x)=\sum_{i=0}^{n-1}{x} * \psi_i^{\text{strict}}
> =\sum_{i=0}^{n-1}{x} * \mathbf{c}^i(\psi)$.
>
> For any $j\in\\{0,1,2,\cdots,n-1\\}$, we have:
>
> - $f_1(\mathbf{c}^j(x))=\sum_{i=0}^{n-1}{\mathbf{c}^j(x)} * \mathbf{c}^i(\psi)$
>
> - $\mathbf{c}^j(f\_1(x))=\mathbf{c}^j(\sum_{i=0}^{n-1}{x} * \mathbf{c}^i(\psi))=\sum_{i=0}^{n-1}{\mathbf{c}^j(x)} * \mathbf{c}^{i+j}(\psi)
> =\sum_{i=0}^{n-1}{\mathbf{c}^j(x)} * \mathbf{c}^{(i+j)\~\text{mod}\~n}(\psi)=\sum_{i=0}^{n-1}{\mathbf{c}^j(x)} * \mathbf{c}^i(\psi)$.
>
> Therefore, $f\_1(\mathbf{c}^j(x))=\mathbf{c}^j(f_1(x))$. According to Eq. (1)  in the paper, $\mathbf{C}_n$-GConv is equivariant.
>
> **2. $\mathbf{C}_n$-R2GConv (Ours) is a relaxed rotation-equivariant block, proven as follows:**
>
> Given input $x$, initial weight $\psi$, an affine transformation function $\mathbf{t}$
> and the learnable perturbation $\Delta$, we obtain the relaxed rotation-equivariant filter
> $\psi_i^{\text{relaxed}}=\mathbf{t}^i(\psi,\Delta)$ in the $i$-order of $\mathbf{C}\_n$.
> Then Cn-R2GConv can be defined as:
> $f_2(x)=\sum_{i=0}^{n-1}{x} * \psi_i^{\text{relaxed}}=\sum_{i=0}^{n-1}{x} * \mathbf{c}^i(\psi,\Delta)$.
>
> For any $j\in\\{0,1,2,\cdots,n-1\\}$, we have:
> - $f_2(\mathbf{c}^j(x))=\sum_{i=0}^{n-1}{\mathbf{c}^j(x)} * \mathbf{t}^i(\psi,\Delta)$
>
> - $\mathbf{c}^j(f_2(x))=\mathbf{c}^j(\sum_{i=0}^{n-1}{x} * \mathbf{t}^i(\psi,\Delta))=\sum_{i=0}^{n-1}{\mathbf{c}^j(x)} * \mathbf{c}^j(\mathbf{t}^i(\psi,\Delta))$.
>
> Therefore, $||f_2(\mathbf{c}^j(x)) - \mathbf{c}^j(f_2(x))|| = ||\sum_{i=0}^{n-1}{\mathbf{c}^j(x)} * \mathbf{t}^i(\psi,\Delta) - \sum_{i=0}^{n-1}{\mathbf{c}^j(x)} * \mathbf{c}^j(\mathbf{t}^i(\psi,\Delta))||
> =||\sum_{i=0}^{n-1}{\mathbf{c}^j(x)} * (\mathbf{t}^i(\psi,\Delta)-\mathbf{c}^j(\mathbf{t}^i(\psi,\Delta)))|| \le \epsilon$.
> According to Eq. (4) in the latest revised version, $\mathbf{C}_n$-R2GConv is a relaxed rotation-equivariant block.
> In particular, when $\Delta=0$, we have $\mathbf{t}^i(\psi,\Delta)=\mathbf{c}^i(\psi)$, thus $||f_2(\mathbf{c}^j(x)) - \mathbf{c}^j(f_2(x))||=0$,
> i.e., $\epsilon=0$. Therefore, $f_2$ is strict rotation-equivariant when $\Delta=0$.
>
> In conclusion, we have mathematically proved that **$\mathbf{C}_n$-R2GConv is a relaxed rotation-equivariant block.**
> We have included this proof in **Appendix A.10** of the latest revised version.

---

> ### Author Response · Authors · 2024-11-17
> **Author Rebuttal to Reviewer AppE: Part 3**
>
> > **Weakness 2**
> The paper is not easy to read, mainly because of: the abundant use of abbreviations and acronyms (ENN, GConv, RRE, SRE, NRE, R2Filter, R2Lift, R2GConv, DR2GConv, PR2GConv, ER2GConv, ER2GCBA, ...), multiple typos ("Rotationa-Equivariant" line 25, "we further exploring" line 62, "More analysis can refer to" line 167, "converge when 66-epoch" and "converges at about 198 in the epoch" lines 401-402, ...) and over-loaded illustrations (e.g., Figure 2 and 3).
>
> **Answer to Weakness 2:**
> Sorry for the confusion.
>
> Thank you very much for your suggestion, which will contribute to the quality of our paper.
> In the latest revised version, we have adopted your suggestion to redefine these concepts (e.g., R2Lift, R2GConv, DR2GConv, PR2GConv, ER2GConv, ER2GCBA,...), and redrawn Figures 2 and 3.
>
> We have carefully revised these typos and thoroughly examined other typos.
> You can check the latest revised version, and the blue color represents the content we have modified or added.
>
> > **Question 1.1:**
> The performance gap between different n-norder cyclic rotation groups is substantial (Table 3).
> What is the reason for such a gap? Is it only due to the "newer equivariant angles"?
>
> **Answer to Question 1.1:**
> R2GConv (or GConv) uses shared convolutional filters through rotational transformations, enabling it to capture rotation-equivariant features. It can learn these features at different rotation angles to enhance the model's robustness.
>
> In Table 3 of the paper, the performance gap between different orders of cyclic rotation groups is primarily due to the introduction of new equivariant angles.
> These additional equivariant angles can offer richer rotation-equivariant features, thereby improving the model's accuracy in detection tasks.
>
> Although as $n$ increases on the group, the model captures more diverse rotation-equivariant features, enhancing performance, this improvement should have an upper limit, with a substantial increase in parameters.
> Generally, the $\mathbf{C}_4$ group provides the optimal balance between performance and parameter efficiency.
>
> > **Question 1.2:**
> How would the introduced architecture perform without equivariance?
>
> **Answer to Question 1.2:**
> We replace all R2GConvs including any variants in R2Det with vanilla Convs while keeping all model parameters consistent. The experimental results on the VOC dataset are as follows:
> |Type|$\text{AP}_{50}$|$\text{AP}_{50:90}$|Params.|
> |-|-|-|-|
> |w/o equivariance|78.6|57.7|3.2M|
> |w/ $\mathbf{C_4}$ strict rotation-equivariance (SRE)|82.9|64.2|2.6M|
> |w/ $\mathbf{C_4}$ relaxed rotation-equivariance (RRE, Ours)|**84.1**|**65.1**|2.6M|
>
> From the table, it is evident that without $\mathbf{C}_4$ (relaxed or strict) rotation-equivariance, the model's performance significantly decreases. This is mainly because vanilla Conv does not possess rotation-equivariance, thus failing to capture objects' rotation-equivariance.
> This also indicates that the performance of our R2Det model is not caused by the introduced architecture.
>
> > **Question 1.3:**
> Also, is there an intuitive explanation for why it is beneficial for 2D object detection (with non-rotating bounding boxes, like on COCO)
> to use C8 instead of C4, despite the output (bounding boxes) only having symmetries in the C4 group?
>
> **Answer to Question 1.3:**
> In our R2Det, we utilize the YOLOv8 detection head, where the output (bounding boxes) is generated through the following steps:
> - Classifying image pixel points for prediction to select candidate points.
> - Generating multiple detection boxes (vertical or horizontal) based on these candidate points.
> - Applying non-maximum suppression (NMS) to eliminate overlapping regions, thereby obtaining the detection results.
>
> During the feature extracting process, using $\mathbf{C_8}$ instead of $\mathbf{C_4}$ can provide finer-grained orientation sensitivity, improved feature representation, better capture of contextual information, robustness to rotation variations, and more effective training dynamics.
>
> Since the detection head is **anchor-free** and **does not directly involve concepts of symmetry or equivariance in the output (bounding boxes)**, the primary focus of the detection process lies in the classification of image pixel points and the application of Non-Maximum Suppression (NMS) based on the image features.
>
> The features derived from the $\mathbf{C_8}$ group bring richer geometric information about objects in images compared to the $\mathbf{C_4}$ group to the detection head of the model, which finally contributes to more accurate and robust object detection.
>
> In fact, our work has always been exploring (relaxed or strict) rotation-equivariant properties on natural image datasets, to promote the performance of **a universal object detection algorithm (e.g., YOLO series, DETR series) that utilizes these properties.**

---

> ### Author Response · Authors · 2024-11-17
> **Author Rebuttal to Reviewer AppE: Part 4**
>
> > **Question 2:** Experiments showed that on both PASCAL VOC and COCO, performance with strict equivariance is inferior to that with the approximate equivariance introduced in the paper. For intuition, could the authors provide specific (intuitive or experimental) examples of 2D object detection in which it is beneficial to break the strict rotation equivariance?
>
> **Answer to Question 2:**
> Thank you for your suggestions, which will help improve the quality of our paper again.
>
> We have adopted your suggestions in the latest revised version and added a 2D object detection case in **Section 4.3** of the paper to illustrate the advantages of relaxed rotation-equivariance.
>
> Finally, we aim to prove that by allowing some flexibility in the equivariance constraints, our model can better capture the anisotropic nature of objects and their contexts, leading to improved performance in various 2D object detection tasks, finally reducing false positives and improving the overall detection accuracy. You can refer to the latest revised version for this example.
>
>
> > **Details Of Ethics Concerns:**
> Potential (but unlikely) plagiarism: the submission is very similar to a preprint that has been on arXiv [1] since August 2024, and Figure 8 even mistakenly re-uses the method name from the ArXiv paper. It would be useful to verify that the authors of the ICLR paper and arXiv paper are the same, which would mean that they just changed their own paper title and method name.
>
> **Response to Details Of Ethics Concerns:** In fact, the current submitted paper ("R2Det: Exploring Relaxed Rotation Equivariance in 2D Object Detection") is our original work and does not violate any ICLR principles. We have clarified this with PC/SPC/AC, and if you have any questions, please feel free to contact with them.
>
> **Thank you again for your review. We look forward to your valuable and timely response, and we are willing to address all your concerns.**

---

> ### Author Response · Authors · 2024-11-20
> **Kind Reminder to Reviewer AppE**
>
> Dear Reviewer AppE,
>
> Thank you for your contribution to the review process of the ICLR25 community.
>
> Since we have earnestly addressed your concerns in our rebuttal responses, we look forward to your valuable and timely responses, which are very important to this work.
>
> If you have any other questions, please don't hesitate to contact us anytime.
>
> Best regards,
>
> Authors.

---

> ### Comment · Reviewer_AppE · 2024-11-20
>
> Dear authors,
>
> Thank you for your detailed answer to the review.
> Since the updated version and the rebuttal:
> - provided additional evidence about the significance of the approach by giving a mathematical proof that $C_n$-R2GConv is a relaxed rotation-equivariant block, and by running an ablation study to show that the performance is not only due to the introduced architecture but also to rotation equivariance itself (in the answer to Question 1.2)
> - fixed the partially incorrect definition and clarified my misunderstanding about whether or not the method was a case of relaxed rotation equivariance, and improved the preliminary section accordingly
> - simplified the reading of the paper by reducing the number of new abbreviations and acronyms, providing easier-to-read illustrations (see Figures 2 and 3), and providing more intuition (in "Visualization of rotational Symmetry-Breaking", lines 517-530)
> - scaled the approach to a "large" version to demonstrate an even more impressive performance
>
> I am willing to raise my rating significantly and recommend acceptance of the paper.
>
> In addition, I have a final question regarding the method: how fast is the approach at inference (in FPS)? Is it faster than YOLO-like models thanks to the reduced number of FLOPS or is it slow because of the "custom" operations? (Even if it is slower, I believe this is not necessarily a disadvantage of the method as it may still be optimized with more fine-tuned implementations)

---

> > ### Author Response · Authors · 2024-11-20
> >
> > Dear Reviewer AppE,
> >
> > Firstly, thank you very much for your suggestions on the revision of our paper, which has contributed to its high quality.
> >
> > Secondly, regarding the concern of R2Det's inference speed, we hope the following explanation can solve your confusion.
> >
> > Indeed, as you mentioned, "custom" operations are causing R2Det's inference speed to be slower.
> > We test the speed of R2Det-N compared with YOLOv8-N on an RTX4090 and the results are as follows:
> >
> > **Table 1:** Inference of R2Det-N and YOLOv8-N per image on an RTX4090.
> > |Model|Inference time|
> > |-|-|
> > |YOLOv8-N|0.9ms|
> > |R2Det-N|2.1ms|
> >
> > The main reason for the slow inference of the model is that the specific **affine transformation** currently does not have a dedicated optimization algorithm.
> >
> > For this purpose, we have modified a dedicated cuDNN operator into Efficient R2GConv on GitHub [1], which has the following model inference speed.
> >
> > **Table 1:** Running Time of Efficient R2GConv with input channel 512, output
> > channel 512, height 640, width 640, kernel size 3, stride 1, and
> > padding 1 on Pytorch and dedicated cuDNN on an RTX4090.
> > |Type|Inference Time|
> > |-|-|
> > |Efficient R2GConv with Pytorch| 0.1431ms|
> > |Efficient R2GConv with a dedicated cuDNN|0.0903ms (-0.0528ms, **36.9\% $\downarrow$)**|
> >
> > In fact, we believe that through specialized operators, inference time can be greatly optimized. **This seems to be an engineering problem that will be solved in the future**.
> >
> > Thirdly, the reason why Efficient R2GConv has lower FLOPs is mainly due to the extensive use of Point-wise and Depth-wise operators. However, Pytorch has low efficiency for the Depth-wise operator, **which is also a direction for improving speed in the future**.
> >
> > Additionally, you can refer to **Appendix A.9** for theoretical calculations of the parameter of our Efficient R2GConv.
> >
> > Finally, thank you for your diligent review again. If you have any questions, we are still happy to answer them.
> >
> > Best regards,
> >
> > Authors.
> >
> > [1] https://github.com/diningeachox/G-CNN

---

### Official Review · Reviewer_L1nv · 2024-11-03

**Soundness:** 3
**Presentation:** 3
**Contribution:** 3
**Rating:** 6
**Confidence:** 3

**Summary:**

This paper points out that symmetry breaking often occurs in the real world. However, traditional GConv-based methods are limited by strict operational rules in group space, ensuring strict equivariance of features only under a limited set of group transformations, making them difficult to adapt. The paper defines the relaxed rotation-equivariant group R4 based on the strict rotation-equivariant group C4 and proposes the relaxed rotation GConv (R2GConv). The paper constructs R2Det using GConv and its derived convolutional structures, achieving excellent results on the PASCAL VOC and MS COCO 2017 datasets, and also verifies the good performance of R2Det in classification and segmentation tasks.

**Strengths:**

1. The paper introduces a novel relaxed rotation-equivariant group convolution (R2GConv), which extends existing equivariant neural networks (ENNs). Additionally, the resulting model, R2Det, shows strong performance across various datasets and tasks.

2. The authors enhance the R2GConv module by incorporating depth-wise and point-wise convolution, and conduct extensive comparative and ablation experiments to confirm its positive impact on the outcomes.

3. The paper is well-written and easy to understand.

**Weaknesses:**

1. Dataset Limitation: The selected datasets, PASCAL VOC and MS COCO 2017, do not emphasize rotation characteristics, which reduces the impact and relevance of the experimental results. To better highlight the effects of SRE and RRE modeling, rotation-specific object detection datasets should be used.

2. Insufficient Baseline Comparison: it would be beneficial to include comparisons with established models in rotation object detection, such as ReDet and FRED, to strengthen the evaluation and provide more convincing evidence.

**Questions:**

1. In Figure 1 on the left, the input in the lower left corner mapped to the output in the upper right corner should use the relaxed rotation-equivariant function.

---

> ### Author Response · Authors · 2024-11-17
> **Author Rebuttal to Reviewer L1nv: Part 1**
>
> **Thank you for taking the valuable time to review the manuscript. The following may address your concerns.**
>
> > **Weakness 1:**
> Dataset Limitation: The selected datasets, PASCAL VOC and MS COCO 2017, do not emphasize rotation characteristics, which reduces the impact and relevance of the experimental results. To better highlight the effects of SRE and RRE modeling, rotation-specific object detection datasets should be used.
>
> **Answer 1:** The selected datasets, PASCAL VOC and MS COCO 2017, are widely recognized and established benchmarks in the field of object detection. Although these datasets do not specifically emphasize rotation characteristics, **they present a more challenging scenario as real-world data often encounters noise or occlusions.**
>
> In fact, our work focuses on exploring rotation-equivariant properties on natural image datasets, in order to propose a universal object detection algorithm (e.g., YOLO series, DETR series) that utilizes these properties. Using rotation-equivariance can learn the intrinsic rotational symmetry of objects, in order to obtain better representations and improve the performance and generalization ability of the model. Therefore, we don't need the dataset to have rotational features.
>
> To further address your concerns, we have conducted experiments on rotated datasets.
> Please refer to the classification experiment on the ROT-MNIST dataset in **Appendix A.5** of the paper.
>
> Our designed ROT-MNIST dataset differs from the standard Rotated MNIST, as it is specifically crafted to assess the robustness of our R2Net. We manipulated the training set by randomly rotating 60,000 images by 0, 90, 180, and 270 degrees, while keeping 10,000 images unaltered in the test set to evaluate the model's performance under rotation.
>
> We compared the training accuracy of YOLOv8-N-CLS and our R2Net-N on the ROT-MNIST dataset. Our R2Net-N demonstrates superior stability and achieves higher accuracy compared to YOLOv8-N-CLS.
> This experiment also demonstrates that our approach can effectively model rotated datasets.
>
>
> > **Weakness 2:**
> Insufficient Baseline Comparison: it would be beneficial to include comparisons with established models in rotation object detection, such as ReDet and FRED, to strengthen the evaluation and provide more convincing evidence.
>
> **Answer 2:**
> Although these approaches involve the concept of rotational equivariance, they still differ significantly in research objectives and tasks, and are different from our 2D detection tasks.
>
> Our R2Det's task is general object detection in natural images, where the output consists of vertical or horizontal bounding boxes.
>
> The reason for our task involving rotation-equivariance is to explore the inherent rotation-equivariant (symmetric) properties of the objects themselves, thereby enhancing model rotational feature learning.
>
> However, ReDet and FRED both focus on a specific task: oriented object detection in the aerial image field, where the output consists of rotated bounding boxes.
> ReDet and FRED focus on rotation-equivariance to better extract rotation-equivariant features, improving the accuracy of orientation prediction.
>
> Therefore, while R2Det and ReDet/FRED both involve the concept of rotation-equivariance, they focus on different tasks.
>
> It is worth noting that, to our knowledge, we are the first to explore (relaxed or strict) rotation-equivariance in the context of natural image object detection tasks.

---

> ### Author Response · Authors · 2024-11-17
> **Author Rebuttal to Reviewer L1nv: Part 2**
>
> > **Question:**
> In Figure 1 on the left, the input in the lower left corner mapped to the output in the upper right corner should use the relaxed rotation-equivariant function.
>
> **Answer 3:** In fact, it is correct to map low symmetry to high symmetry using rotation-equivariant functions.
>
> Define the function $Sym(\cdot)$, which denotes the degree of symmetry of $\cdot$, and
> the strict equivariant function $f_{\text{strict}}$ with input $x$.
>
> According to the **Curie principle** [1], we have the following conclusion:
> $Sym(x) \leq Sym(f_{\text{strict}}(x))$.
>
> Therefore, the strict equivariant function cannot map the input to lower symmetry since its output should at least have the same symmetry as the input, which is known as the Symmetry-Breaking problem.
>
> On the other hand, the strict equivariant function can map the input to higher symmetry, as the input, after going through the strict equivariant function, can exhibit equivariance, thus achieving higher symmetry.
>
> Due to the lower symmetry present in real-world data, modeling it using a strictly equivariant function fails to map the input to lower symmetry, thereby deviating from the characteristics of real-world data and affecting feature learning.
>
> For more details, please refer to Figure 1 in (Kaba and Ravanbakhsh, 2023) [1].
>
> Conversely, the relaxed equivariant function can resolve the Symmetry-Breaking problem.
> The rationale is simple: using the relaxed equivariant function can relax the original symmetry of the input,
> thereby mapping it to lower symmetry, aligning it with the characteristics of real-world data, and enhancing feature learning.
>
>
> **References:**
>
> [1] Sekou-Oumar Kaba and Siamak Ravanbakhsh. Symmetry breaking and equivariant neural networks.
>
> **Thank you again for your review. We look forward to your valuable and timely response, and we are willing to address all your concerns.**

---

> ### Author Response · Authors · 2024-11-20
> **Kind Reminder to Reviewer L1nv**
>
> Dear Reviewer L1nv,
>
> Thank you for your contribution to the review process of the ICLR25 community.
>
> Since we have earnestly addressed your concerns in our rebuttal responses, we look forward to your valuable and timely responses, which are very important to this work.
>
> If you have any other questions, please don't hesitate to contact us anytime.
>
> Best regards,
>
> Authors.

---

> ### Author Response · Authors · 2024-11-24
> **Looking forward to your valuable reply**
>
> Dear Reviewer L1nv,
>
> Sorry to bother you again.
>
> As the rebuttal phase nears the end, we would like to know if we have addressed your concerns.
>
> If you have any remaining concerns, please let us know. We look forward to your valuable reply.
>
> Thank you for your efforts in our paper.
>
> Best regards,
>
> Authors.

---

> > ### Comment · Reviewer_L1nv · 2024-12-02
> >
> > Dear authors, thank you for your thorough responses, which addressed my concerns, I maintain my initial score.

---

> > > ### Author Response · Authors · 2024-12-02
> > >
> > > Dear Reviewer L1nv,
> > >
> > > We are delighted to hear that your concerns have been thoroughly addressed. Thank you for your efforts in reviewing the manuscript and for recognizing our work.
> > >
> > > Best regards,
> > >
> > > Authors.

---

### Official Review · Reviewer_e3oq · 2024-11-03

**Soundness:** 2
**Presentation:** 2
**Contribution:** 2
**Rating:** 6
**Confidence:** 5

**Summary:**

This paper introduces R2Det, a novel object detection model that explores Relaxed Rotation-Equivariance (RRE) to handle real-world scenarios where strict rotational symmetries are often violated. RRE is incorporated into group convolution by introducing the Relaxed Rotation-Equivariant Filter (R2EFilter) and the Relaxed Rotation-Equivariant Group Convolution (R2GConv). The paper further proposes R2Net for image feature extraction and R2Det for 2D object detection, achieving improved convergence and performance with fewer parameters.

**Strengths:**

This work makes a valuable contribution to the field of object detection by addressing the limitations of traditional, strictly rotation-equivariant models and exploring the potential of RRE through ER2GConv.
This work presents a significant contribution to the field of object detection by addressing a crucial limitation in handling real-world scenarios, where strict rotational symmetries are rarely observed. The authors introduce a novel approach, Relaxed Rotation-Equivariance (RRE), which effectively addresses this limitation. The proposed R2Det model leverages RRE to achieve remarkable performance with a significantly reduced parameter count compared to other leading models. This showcases the model's efficiency and its ability to achieve high accuracy while requiring fewer computational resources. The paper further strengthens its arguments with a rigorous mathematical framework, providing theoretical underpinnings for RRE's effectiveness. Moreover, the plug-and-play nature of the proposed ER2GConv layer allows for seamless integration into existing object detection models, making it a versatile and readily applicable technique. This combination of novelty, theoretical soundness, efficiency, and integration capabilities makes this research highly valuable for advancing the field of object detection.
However, the paper lacks sufficient exploration of the key parameter b, which controls the perturbation factor Δ. While the paper mentions that b=0.1 yields the best results, more extensive experiments with intermediate values of b would strengthen the argument for the necessity of this perturbation parameter and provide a deeper understanding of its influence on performance. Additionally, it would be beneficial to investigate the performance of R2Det in larger configurations, such as “R2Det-L”, and compare its performance to the corresponding “Large” versions of YOLO models. This would provide a more complete assessment of the model's scalability and potential limitations in handling more complex and computationally demanding tasks. The clarity and persuasiveness of the paper can be further enhanced by addressing these specific concerns, and by providing concise explanations for abbreviations like SO(2) (Special Orthogonal Group in 2D space) and ER2GCBA (Efficient Relaxed Rotation-Equivariant Group Convolution plus -BA, where -BA likely refers to a specific architectural component or technique).

* This work makes a valuable contribution to the field of object detection by addressing the limitations of traditional, strictly rotation-equivariant models and exploring the potential of RRE through ER2GConv.
* The authors introduce a novel approach, Relaxed Rotation-Equivariance (RRE), which effectively addresses the limitation of handling real-world scenarios where strict rotational symmetries are rarely observed.
* The proposed R2Det model leverages RRE to achieve remarkable performance with a significantly reduced parameter count compared to other leading models, showcasing the model's efficiency and ability to achieve high accuracy while requiring fewer computational resources.
* The paper further strengthens its arguments with a rigorous mathematical framework, providing theoretical underpinnings for RRE's effectiveness.
* The plug-and-play nature of the proposed ER2GConv layer allows for seamless integration into existing object detection models, making it a versatile and readily applicable technique.

**Weaknesses:**

* The paper lacks sufficient exploration of the key parameter b, which controls the perturbation factor Δ, and more extensive experiments with intermediate values of b would strengthen the argument for the necessity of this perturbation parameter.
** In page 8, Figure 4(a) and Table 1, the results presented demonstrate a minimal improvement in AP when (b=0.1) compared to (b=0).
** Performance deteriorates when (b>0.1), and it would be beneficial to conduct more thorough experiments, especially within the interval [0, 0.1], e.g., 0.02, 0.05, to provide a more definitive analysis of the value of b and its impact on the model’s performance.
* It would be beneficial to provide concise explanations for abbreviations like SO(2) (Special Orthogonal Group in 2D space) and ER2GCBA (Efficient Relaxed Rotation-Equivariant Group Convolution plus -BA, where the meaning of -BA remains undefined).
* It would be beneficial to investigate the performance of R2Det in larger configurations, such as 'R2Det-L', and compare its performance to the corresponding 'large' versions of YOLO models, to provide a more complete assessment of the model's scalability and potential limitations in handling more complex and computationally demanding tasks.
* It would be beneficial to include a comparative study with more recent and advanced object detection models, such as YOLOv11 and other models, to provide a broader context and demonstrate the model's performance relative to the state-of-the-art.
* I noticed an interesting discrepancy in the results presented on page 8, specifically in Table 1 and Table 2. While both tables use the VOC test dataset, the reported AP scores for the SRE (b=0) model in Table 1 (C4) differ from the reported SRE scores in Table 2 (C4).
** In Table 1, the AP50(%) and AP50:95(%) for b=0 are 83.8 and 64.4 respectively, whereas in Table 2, the SRE AP50(%) and AP50:95(%) for C4 are 82.9 and 64.2.
** This discrepancy raises a question regarding potential differences in the implementation of SRE versus the b=0 setting, or if it could be due to variations in the experimental runs. The authors may please comment on this interesting observation and clarify the reasons behind the difference in AP scores.

**Questions:**

* The paper lacks sufficient exploration of the key parameter b, which controls the perturbation factor Δ, and more extensive experiments with intermediate values of b would strengthen the argument for the necessity of this perturbation parameter.
* Performance deteriorates when (b>0.1), and it would be beneficial to conduct more thorough experiments, especially within the interval [0, 0.1], e.g., 0.02, 0.05, to provide a more definitive analysis of the value of b and its impact on the model’s performance.
* It would be beneficial to provide concise explanations for abbreviations like SO(2) (Special Orthogonal Group in 2D space) and ER2GCBA (Efficient Relaxed Rotation-Equivariant Group Convolution plus -BA, where the meaning of -BA remains undefined).
* It would be beneficial to investigate the performance of R2Det in larger configurations, such as 'R2Det-L', and compare its performance to the corresponding 'large' versions of YOLO models, to provide a more complete assessment of the model's scalability and potential limitations in handling more complex and computationally demanding tasks.
* It would be beneficial to include a comparative study with more recent and advanced object detection models, such as YOLOv11 and other models, to provide a broader context and demonstrate the model's performance relative to the state-of-the-art.
* This discrepancy raises a question regarding potential differences in the implementation of SRE versus the b=0 setting, or if it could be due to variations in the experimental runs. The authors may please comment on this interesting observation and clarify the reasons behind the difference in AP scores.

---

> ### Author Response · Authors · 2024-11-17
> **Author Rebuttal to Reviewer e3oq: Part 1**
>
> **Thank you for your appreciation of our work and for taking the valuable time to review the manuscript. The following may address your concerns.**
>
> >**Weakness 1:**
> The paper lacks sufficient exploration of the key parameter b, which controls the perturbation factor Δ, and more extensive experiments with intermediate values of b would strengthen the argument for the necessity of this perturbation parameter. **In page 8, Figure 4(a) and Table 1, the results presented demonstrate a minimal improvement in AP when (b=0.1) compared to (b=0).** Performance deteriorates when (b>0.1), and it would be beneficial to conduct more thorough experiments, especially within the interval [0, 0.1], e.g., 0.02, 0.05, to provide a more definitive analysis of the value of b and its impact on the model's performance.
>
> **Answer:**
> Thank you very much for your suggestions. We conduct more comprehensive experiments regarding the hyperparameter $b$, and the results are as follows:
>
> |$b$|$\text{AP}_{50}$|$\text{AP}_{50:95}$|
> |-|-|-|
> |0|83.8|64.4|
> |0.01|84.0|64.9|
> |0.02|84.2|65.2|
> |0.04|84.3|65.5|
> |**0.06**|**84.3**|**65.6**|
> |0.08|84.2|65.3|
> |0.1|84.1|65.1|
> |0.2|83.5|64.3|
> |0.4|83.6|64.4|
> |0.6|82.4|62.6|
> |0.8|80.7|59.7|
>
> **First of all, we need to emphasize that $b$ is a hyperparameter that sets a Uniform distribution $\mathcal{U}(-b,b)$ for $\Delta$. In fact, the RRE ($b=0$) model is not equivalent to the SRE model.**
>
> When $b=0$, all initial values of $\Delta$ are $0$. At this moment, the RRE ($b=0$) model is equivalent to the SRE model, but $\Delta$ still undergoes end-to-end updates with gradient descent. After that moment, the RRE ($b=0$) model is not equivalent to the SRE model.
>
> For this experiment, we intend to explore the impact of the initial values of $\Delta$ on the model performance.
>
> From the table, it can be seen that when $b\in[0.01, 0.1]$, the model's performance is higher than that of $b=0$, and they all converge to similar results, especially when $b=0.06$, the model performance is the highest.
>
> When $b=0$, meaning all initial values of $\Delta$ are set to $0$, the updates of $\Delta$ lack an initial push for convergence, resulting in the model being lower than models with a small initial push.
>
> When $b=0.2$ or $b=0.4$, the model performance starts to decrease, and when $b=0.6$ or $0.8$, the model performance decreases significantly. It can be inferred that when $b>0.1$ and as $b$ increases, the model performance will decrease significantly. We speculate that excessive disturbance may lead to the model updating towards incorrect gradients, resulting in decreased model performance.
>
> From the above experiments, we can conclusions:
> The initial values of $\Delta$ have a significant impact on the model. Providing $\Delta$ with small initial values (such as $b=0.02, 0.04, 0.06$) is beneficial for the model to converge better. However, providing $\Delta$ with large initial values (such as $b=0.6, 0.8$) is not conducive to better convergence of the model. This conclusion can be found in Figure 4(a) of the paper.

---

> ### Author Response · Authors · 2024-11-17
> **Author Rebuttal to Reviewer e3oq: Part 2**
>
> > **Weakness 2:**
> It would be beneficial to provide concise explanations for abbreviations like SO(2) (Special Orthogonal Group in 2D space) and ER2GCBA (Efficient Relaxed Rotation-Equivariant Group Convolution plus -BA, where the meaning of -BA remains undefined).
>
> **Answer 2:**
> Sorry for the confusion.
>
> The SO(2) group is an infinite group that contains a set of all two-dimensional rotation angles.
> In fact, $\mathbf{C}_n$ is a discrete subgroup of SO(2).
> The '-BA' denotes BatchNorm and Activation operation, which is a standard practice in convolutional neural networks.
>
> Thank you very much for pointing out these issues. In the latest revised version, we have removed the notation '-BA' and provided detailed explanations of SO(2). Furthermore, **Reviewer AppE** has provided valuable suggestions regarding these abbreviations, and we have adopted a standardized notation. These suggestions have contributed to the high quality of the paper. Please see the latest revised version.
>
> > **Weakness 3:**
> It would be beneficial to investigate the performance of R2Det in larger configurations, such as 'R2Det-L', and compare its performance to the corresponding 'large' versions of YOLO models, to provide a more complete assessment of the model's scalability and potential limitations in handling more complex and computationally demanding tasks.
>
> **Answer 3:**
> We provide the results of R2Det-L on VOC and COCO datasets, as shown below:
> |Model|Year|Dataset|$\text{AP}_{50}$|$\text{AP}_{50:95}$|FLOPs|Params.|
> |-|-|-|-|-|-|-|
> |RT-DETRv2-L|2024|COCO|71.6|53.4|136G|42M|
> |YOLO11-L|2024|COCO|70.1|53.4|86.9G|25.3M|
> |YOLO11-X|2024|COCO|71.6|54.7|194.9G|56.9M|
> |R2Det-L|2024|COCO|**72.4**|**56.1**|28.3G|42.8M|
> |R2Det-L|2024|VOC|88.3|71.9|28.1G|42.7M|
>
> The table shows that R2Det-L achieves **state-of-the-art performance** in the COCO dataset compared to the latest models. We will add this latest result and comparison in the final version.
>
> > **Weakness 4:**
> It would be beneficial to include a comparative study with more recent and advanced object detection models, such as YOLOv11 and others, to provide a broader context and demonstrate the model's performance relative to the state-of-the-art.
>
> **Answer 4:**
> In fact, YOLO11 was only released on **September 27, 2024**, and we did not notice this latest model. You can view the latest results and comparisons in **Answer 3**.
>
> > **Weakness 5:**
> I noticed an interesting discrepancy in the results presented on page 8, specifically in Table 1 and Table 2. While both tables use the VOC test dataset, the reported AP scores for the SRE (b=0) model in Table 1 (C4) differ from the reported SRE scores in Table 2 (C4). **In Table 1, the AP50(%) and AP50:95(%) for b=0 are 83.8 and 64.4 respectively, whereas in Table 2, the SRE AP50(%) and AP50:95(%) for C4 are 82.9 and 64.2.** This discrepancy raises a question regarding potential differences in the implementation of SRE versus the b=0 setting, or if it could be due to variations in the experimental runs. The authors may please comment on this interesting observation and clarify the reasons behind the difference in AP scores.
>
>
> **Answer 5:** In fact, the model with $b=0$ in Table 1 corresponds to the RRE ($b=0$) model, not the SRE model in Table 2. The initial value of $\Delta$ in the RRE ($b=0$) model is $0$, but it updates end-to-end with gradient descent. Therefore, they have different reported results.
>
> For a detailed explanation, please see the explanation in **Weakness 1**.
>
> > **Answer to Questions part**: See the answers in **Weaknesses** part.
>
> **Thank you again for your review. We look forward to your valuable and timely response, and we are willing to address all your concerns.**

---

> ### Author Response · Authors · 2024-11-20
> **Kind Reminder to Reviewer e3oq**
>
> Dear Reviewer e3oq,
>
> Thank you for your contribution to the review process of the ICLR25 community.
>
> Since we have earnestly addressed your concerns in our rebuttal responses, we look forward to your valuable and timely responses, which are very important to this work.
>
> If you have any other questions, please don't hesitate to contact us anytime.
>
> Best regards,
>
> Authors.

---

> ### Author Response · Authors · 2024-11-24
> **Looking forward to your valuable reply**
>
> Dear Reviewer e3oq,
>
> Sorry to bother you again.
>
> As the rebuttal phase nears the end, we would like to know if we have addressed your concerns.
>
> If you have any remaining concerns, please let us know. We look forward to your valuable reply.
>
> Thank you for your efforts in our paper.
>
> Best regards,
>
> Authors.

---

> > ### Comment · Reviewer_e3oq · 2024-11-29
> >
> > Thanks for the kindly reply. It addressed my concerns.I keep my scores for the acceptance of this paper.

---

> > > ### Author Response · Authors · 2024-11-29
> > >
> > > Dear Reviewer e3oq,
> > >
> > > Thank you very much for your efforts in reviewing the manuscript and for acknowledging our work.
> > >
> > > Best regards,
> > >
> > > Authors.

---

### Author Response · Authors · 2024-11-17
**Welcome to discussion!**

Dear Reviewers:

Thank you for your review! We have uploaded the latest revised version, with blue markings indicating new or modified content. We look forward to your valuable discussion.

Best regards,

Authors.

---

### Meta-Review · Area_Chair_FFqm · 2024-12-18

**Metareview:**

The paper introduces Relaxed Rotation-Equivariant GConv (R2GConv), addressing the limitations of traditional GConv models in handling symmetry-breaking, particularly rotational symmetry-breaking. Traditional GConv assumes strict equivariance, which fails to account for real-world deviations in symmetry. The proposed R2GConv relaxes the rotational transformations, introducing only minimal additional parameters.The authors apply R2GConv in the Relaxed Rotation-Equivariant Network (R2Net), and develop the R2Det object detector for 2D detection tasks. Experimental results demonstrate that R2GConv improves natural image classification, while R2Det achieves strong performance in 2D object detection with better generalization and robustness under symmetry-breaking conditions.

All reviews agree to accept this paper. The authors are required to update this paper when preparing the final version, considering the valuable reviews during the rebuttal period.

**Additional Comments On Reviewer Discussion:**

All reviews agree to accept this paper. The authors are required to update this paper when preparing the final version, considering the valuable reviews during the rebuttal period.

---

### Decision · Program_Chairs · 2025-01-22

Accept (Poster)